# Non-local X-ray intermolecular radiative decay probes solvation shell of ions in water

Johan Söderström [1] ✉, Lucas M. Cornetta [2], Victor Ekholm [3], Vincenzo Carravetta[4], Arnaldo Naves de Brito [5], Ricardo Marinho [6,7], Marcus Agåker [1,3], Takashi Tokushima [3], Conny Såthe[3], Anirudha Ghosh[3], Dana Bloß [8], Andreas Hans [8], Florian Trinter [9], Iyas Ismail [10], Debora Vasconcelos[1], Joel Pinheiro[1], Yi-Ping Chang [11], Manuel Harder[11], Zhong Yin [12], Joseph Nordgren[1], Gunnar Öhrwall [3], Hans Ågren[1,13], Jan-Erik Rubensson [1] & Olle Björneholm[1] ✉

Aqueous solutions are crucial in chemistry, biology, environmental science, and technology. The chemistry of solutes is influenced by the surrounding solvation shell of water molecules, which have different chemical properties than bulk water due to their different electronic and geometric structure. It is experimentally challenging to selectively investigate this property-determining electronic and geometric structure. Here, we report experimental results on the non-local X-ray emission process Intermolecular Radiative Decay, for the prototypical ions $Na^+$ and $Mg^{2+}$ in water. We show that, in Intermolecular Radiative Decay, an electron from the solvation shell fills a core hole in the solute, and the released energy is emitted as an X-ray photon. We interpret the underlying mechanism using theoretical calculations, and show how Intermolecular Radiative Decay will allow us to meet the challenge of selectively probing the solvation shell from within.

The fundamental importance of water in chemical reactions is intricately linked to the behavior of water molecules in the immediate vicinity of the reactants[1–3]. These solvation shells are interfaces between the solutes and the bulk solvent. Together with a metal ion, they form weakly bonded metal-aqua complexes of fundamental importance in environmental, biological, and applied chemistry. The structure and dynamics in solvation shells differ substantially from bulk water; for example, the water molecules around metal cations are stronger proton donors than the bulk water molecules[3,4]. In the case of ionic solutes, the water molecules surrounding an ion are oriented with their positive (negative) end towards the anion (cation), and with

increasing ionic charge, the ion-water distance decreases, leading to the ion-water interaction changing from ion-dipole towards dative covalent bonding. The importance of the solvation shell extends beyond small inorganic ions; for example, the functionality of large biomolecules such as proteins is affected by their surrounding hydration shell[5,6].

Several methods are used to determine the molecular geometry of a solvation shell. X-ray and neutron scattering are general methods to measure interatomic distances. They are straightforward to apply to liquids, and they can also provide information about solvation shells[5,7]. The interatomic distance and coordination around specific solutes can

[1]Department of Physics and Astronomy, Uppsala University, Uppsala, Sweden. [2]Instituto de Física, Universidade de São Paulo, São Paulo, Brazil. [3]MAX IV Laboratory, Lund University, Lund, Sweden. [4]Institute of Chemical and Physical Processes, CNR-IPCF, Pisa, Italy. [5]Institute of Physics Gleb Wataghin, State University of Campinas, Campinas, Brazil. [6]Department of Physics, University of Brasilia, Brasilia, Brazil. [7]Institute of Physics, Federal University of Bahia, Salvador, Brazil. [8]Institute of Physics, University of Kassel, Kassel, Germany. [9]Molecular Physics, Fritz-Haber-Institut der Max-Planck-Gesellschaft, Berlin, Germany. [10]Sorbonne Université, CNRS, Laboratoire de Chimie Physique - Matière et Rayonnement, LCPMR, Paris, France. [11]European XFEL, Schenefeld, Germany. [12]International Center for Synchrotron Radiation Innovation Smart, Tohoku University, Sendai, Japan. [13]Faculty of Chemistry, Wroclaw University of Science and Technology, Wroclaw, Poland. ✉e-mail: Johan.Soderstrom@physics.uu.se; Olle.Bjorneholm@physics.uu.se

be inferred from measurements of extended X-ray absorption fine structure (EXAFS)[7,8]. Nuclear magnetic resonance (NMR) and vibrational spectroscopy selectively probe the structure and dynamics around solutes[2,5,9]. While these techniques successfully provide information about the geometry, the investigation of the electronic structure of solvation shells remains an experimental challenge. An experimental probe of the electronic interaction between the solute with the neighboring solvent could thus lead to an increased understanding of this interaction.

X-ray-based spectroscopies are widely used to probe the electronic structure of matter[10]. Still, progress has been hampered by limited selectivity for solvation shells. The element specificity associated with quasi-atomic core holes is insufficient to separate water molecules in the solvation shell from bulk water. In the soft X-ray range, core-hole decay is typically dominated by non-radiative local Auger-Meitner decay, where one valence electron fills the core hole while another is emitted. Non-local decay processes, such as intermolecular Coulombic decay (ICD), are sensitive to local neighbors: the decay of a core hole on a solute involves electrons from neighboring water molecules[11]. In fact, such core ICD processes have been observed for ions, such as $Mg^{2+}$, $Al^{3+}$, $K^+$, and $Ca^{2+}$ in aqueous solutions[12–15]. Very recently, resonant ICD was used to probe the local surroundings of $Ca^{2+}$ in water, including ion pair formation[16]. However, due to the short inelastic mean free path of electrons in water, these ICD electrons can only provide information about the surface region of the solution. In radiative core-hole decay, X-rays are emitted as outer electrons fill atom-specific local core holes, which allows for more bulk-sensitive studies than using ICD electrons. The radiative decay of solvated ions is largely unexplored. Probing non-local radiative processes of solvated ions to gain information about nearest neighbors, in analogy to non-radiative ICD, has not been reported before for any type of liquid system.

The electronic configuration of the free metal ions $Na^+$ and $Mg^{2+}$ is neon-like, $1s^2 2s^2 2p^6$, with the highest occupied states being the 2p core levels; for simplicity, we label these metal ions M of charge q as $M^q$. Photoionization of the M 1s level in such an $M^q$ ion results in an $M^{q+1}$ ion with a 1s core hole. The most energetic radiative decay of the 1s core hole in these $M^{q+1}$ ions is the $K_\alpha$ decay, $M^{q+1} 1s^{-1} \rightarrow M^{q+1} 2p^{-1} + h\nu$. In aqueous solution, $Na^+$ and $Mg^{2+}$ ions are surrounded by a solvation shell, which on average consists of six water molecules in an octahedral arrangement at an average ion-oxygen distance of 2.43 and 2.07 Å, respectively[17]. For the solvated $Na^+$ and $Mg^{2+}$ ions, any radiative decay at higher photon energy than $K_\alpha$ would thus indicate a process involving electrons from the surrounding water molecules. Here, we demonstrate, by measurements and theoretical calculations, how an electron from a neighboring water molecule fills the M 1s core hole, and the released energy is emitted as an X-ray photon, as schematically

shown in Fig. 1. The non-local character of the intermolecular radiative decay (IRD) enables unique investigations of the solvation shell.

Radiative transitions involving a core hole on one atom and a valence orbital mainly located on a neighboring atom have been well known for a long time. In transition metal-oxygen ligand systems, such as $MnO_4^-$, $CrO_4^{2-}$, and $VO_4^{3-}$, interatomic radiative decay has been observed from orbitals primarily located on the ligand oxygen atoms into core holes centered on the metal ion, see, e.g., Refs. 18–23. Compared to the ions studied here, which form much weaker bonds with the surrounding water, the systems studied in Refs. 18–23 are held together more strongly either by covalent bonds in molecules, or ionic/covalent bonds in the crystalline systems. These interatomic transitions observed for the strongly bonded molecular systems are still within the same molecule and can therefore be characterized as intramolecular. IRD, on the other hand, is intermolecular, opening up the possibility of probing chemically and biologically relevant systems in an aqueous environment.

In this work, we use the prototypical and well-known $Na^+$ and $Mg^{2+}$ ions in water as showcases to demonstrate the existence of IRD. Moreover, by theoretical modeling of the process, we show how the IRD depends on factors such as the electronic structure, ion-water distances, orientational (dis-)order, and chemical composition of the immediate surroundings. This demonstrates the potential of IRD to probe the solvation shell from within, providing uniquely selective information on how it differs from bulk water. We foresee that IRD can find applications in studies of the interaction between weakly interacting neighbors, much in the same way as ICD in the non-radiative channel.

## Results
### IRD: observation and interpretation

Ionization of the 1s level of either of the $M^q$ metal ions results in an $M^{q+1}$ $1s^{-1}$ state, and in the middle of Fig. 2, we present the X-ray emission spectra from the decay of this core-hole state for solvated $Na^+$ and $Mg^{2+}$ ions. The main spectral peak for both ions is located toward the lower-photon-energy side of the spectra, ~1041 eV for Na and ~1254 eV for Mg. These peaks are due to the $K_\alpha$ decay, $M^{q+1} 1s^{-1} \rightarrow M^{q+1} 2p^{-1} + h\nu$, i.e., a local process, in which the M 1s hole is filled by a M 2p electron.

At somewhat higher photon energy than the $K_\alpha$, around ~1060 eV for Na and ~1295 eV for Mg, there are weaker spectral features, which moreover exhibit substructures. As M 2p is the highest occupied orbital of free $Na^+$ and $Mg^{2+}$ ions, these weaker spectral features must involve the neighboring water molecules. This can schematically be written as $M^{q+1} 1s^{-1} + H_2O \rightarrow M^q + H_2O^+ val^{-1} + h\nu$, where $val^{-1}$ denotes a valence hole on the water molecules in the first solvation shell. Just as the $K_\alpha$ photon energy is determined by the energy difference between the $M 1s^{-1}$ and $M 2p^{-1}$ states, the photon energy of such non-local

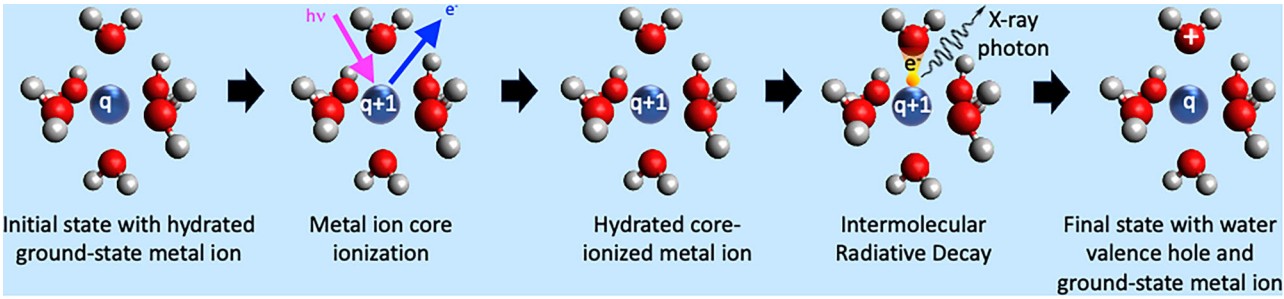

**Fig. 1 | Schematic illustration of the Intermolecular Radiative Decay (IRD) process for a solvated $M^q$ ion.** From the left, we show the ground state with a $M^q$ ion surrounded by six water molecules, followed by core photoionization, which results in a core-ionized $M^{q+1}$ ion. In the IRD process, an electron from a solvation-shell water molecule fills the M core hole, and the released energy is emitted as an X-ray photon. In the final state, the $M^q$ ion is back to its ground state, and the hole is in the valence levels of the solvation-shell water molecules.

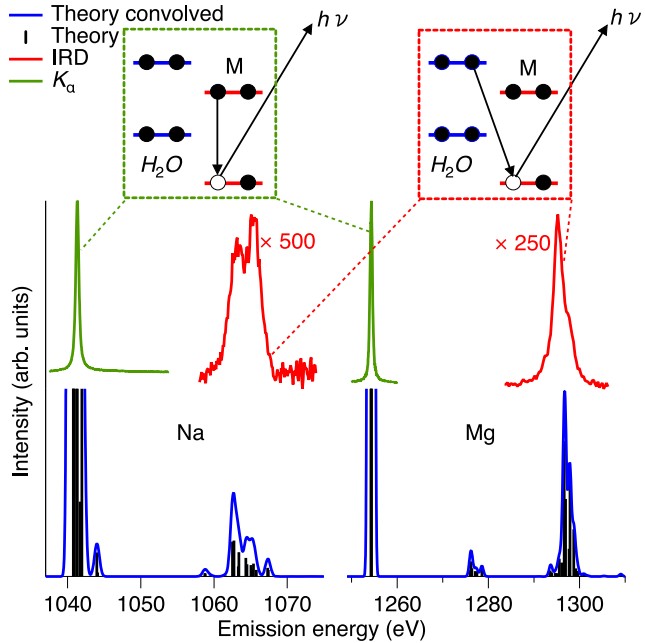

**Fig. 2 | Theoretical and experimental overview of the IRD signal compared to $K_\alpha$.** Top: Schematic illustration of the electronic transitions in the local $K_\alpha$ decay (green box) and non-local IRD decay (red box). Middle: Experimentally measured X-ray emission spectra for Na⁺ and Mg²⁺ after 1s ionization of the metal ion M in the energy range of $K_\alpha$ and IRD. For both ions, the spectra exhibit two main features, of which the most intense one is due to the local $K_\alpha$ emission, $M^{q+1} 1s^{-1} \rightarrow M^{q+1} 2p^{-1} + h\nu$. The weaker features at higher photon energies agree well with the energies estimated for the non-local intermolecular radiative decay IRD, $M^{q+1} 1s^{-1} + H_2O \rightarrow M^q + H_2O^+ val^{-1} + h\nu$. Bottom: Theoretical X-ray emission spectra for Na⁺ and Mg²⁺ calculated for $M^q[H_2O]_6$ clusters. The absolute energy scales of the theoretical spectra are shifted so that the $K_\alpha$ emission aligns with the experimental value. The feature around 1277 eV in the Mg spectrum is due to IRD involving the inner-valence orbital $2a_1$ of water, which mainly consists of O 2s. This will not be further discussed, as the corresponding spectral region was not measured. The theoretical curves were obtained using the Hartree-Fock approximation.

radiative decay processes would similarly be given by the energy difference between the $M^{q+1} 1s^{-1} + H_2O$ and $M^q + H_2O^+ val^{-1}$ states.

In a first approximation, we can use experimentally determined values for $M^{q+1} 1s^{-1}$ from Ref. 14 and $H_2O^+ val^{-1}$ to roughly estimate the photon energy. For bulk liquid water, the $H_2O^+ val^{-1}$ states span from ~10 to ~20 eV binding energy. Assuming that the $H_2O^+ val^{-1}$ states of the solvation shell are not very different, we use 15 eV as an approximate average value for the $H_2O^+ val^{-1}$ energy. This yields about 1062 and 1295 eV for the photon energy for such non-local radiative decays for solvated Na⁺ and Mg²⁺ ions, respectively. These values agree quite well with the photon energies of the spectral features observed at higher photon energies, thereby supporting the assignment.

For an accurate interpretation of the observed features, it is relevant to estimate their intensity as predicted by a theoretical model. We have then considered a number of molecular structures and computational models to predict the X-ray emission spectra of solvated Na⁺ and Mg²⁺ ions; details will be provided in the Theory section and in the Supplementary Information. In the lower panel of Fig. 2, we show the spectra computed for a model consisting of Na⁺ and Mg²⁺ ions within a cluster of six water molecules and using the Hartree-Fock (HF) approximation. In addition to the local $K_\alpha$ decay at lower photon energies, spectral features at higher photon energies are obtained for both ions; they coincide very well with the experimentally observed bands. Moreover, we measured the relative intensity of the IRD line vs. the $K_\alpha$ line of ~1% for Na and ~1.5% for Mg. The calculated IRD/$K_\alpha$

intensity ratios are 1.1 and 1.5%, respectively, i.e., in excellent agreement with the experimental result.

Additionally, an analysis of the involved electronic states, to be further discussed below, reveals that the contributions to the IRD bands are primarily due to decay from hybrid orbitals composed mainly of water and, to a lesser extent, metal ion orbitals.

The IRD process can be seen as a radiative relative of the growing family of non-local decay processes, such as ICD and electron-transfer-mediated decay (ETMD)[11,24]. IRD bears some resemblance to the non-local valence-hole decay process, radiative charge transfer (RCT). This process, in which a photon is emitted when a valence electron is transferred from the neutral environment to a charged species with one or more valence holes, is well-known from ion collision studies, see, e.g., Refs. 25,26. Closer to the present context, RCT has been indirectly inferred to occur for Ne and Ar dimers[27–30], and recently directly observed for homogeneous and heterogeneous rare-gas clusters[31–33]. While RCT has been observed in cases where other decay channels are energetically forbidden, the IRD observed here occurs in competition with non-radiative Auger-Meitner decay and local radiative $K_\alpha$ decay. Unlike the valence-hole decay process, RCT, and IRD involve a core hole, which opens the possibility to use them for chemically selective studies. In order to observe IRD a non-zero overlap between the solute and solvent orbitals is required. Furthermore, in the present systems IRD and $K_\alpha$ are well separated; in other solutes, different energetic overlap may complicate interpretation.

We conclude that the spectral features observed at higher photon energies are due to a non-local radiative decay process, IRD, involving the transfer of an electron from the water molecules in the first solvation shell to the core-ionized metal ion. The calculated spectra agree quite well with the experiments in terms of energy positions of the IRD features and their intensity relative to $K_\alpha$. This gives us confidence to use the calculations in combination with the experimental results to further explore the mechanisms of the IRD decay, how it depends on the surroundings of the ion, and to discuss how IRD can provide an insider's view of the first solvation shell.

## The IRD mechanism

To investigate the mechanisms behind IRD, specifically what influences its intensity and where the electron filling the core hole comes from, we will start by examining a simplified type of model system $M^q[H_2O]_6$. This consists of a metal cation $M^q$ surrounded by six water molecules in a symmetric $D_{2h}$ structure, see Supplementary Fig. 3. In the one-particle picture, the electronic structures of these systems can be described by orbitals, here denoted hybrid orbitals $\psi_\mu$, each of which is built by linear combinations of atomic orbitals (LCAO) $\chi_\nu$. In the usual manner, the atomic orbitals are a set of Cartesian-Gaussian functions centered on the metal ion and on all water-related atoms. For simplicity, the $\chi$ has been indexed simply by $\nu$, in such a way that one can write $\psi_\mu = \sum_\nu C_\mu^\nu \chi_\nu$, where $C_\mu^\nu$ is the hybrid orbital coefficient. The most relevant hybrid orbitals $\psi_\mu$ of the symmetric model are shown in Supplementary Fig. 4.

As a starting point, we will follow the development of the charge distribution from the initial state, over the core-ionized state, to the different final states reached by radiative decay. Figure 3 shows the Mulliken charges of the $M^q[H_2O]_6$ ions in the three states, see also Supplementary Table 1. For a free, isolated Na/Mg ion, the initial state has a charge of +1/+2, and both the core-ionized and the final states have a charge of +2/+3, as indicated in black. For the solvated ions, the total charge develops in the same way, but the presence of the water molecules allows for charge redistribution. In the ground state of the Na⁺[H₂O]₆ ion cluster, the Na ion charge is +0.95, i.e., very close to +1. A small charge of +0.05 is delocalized onto the six water molecules, which means that the Na⁺–water interaction is quite close to an ideal ion-dipole interaction. For the Mg²⁺[H₂O]₆ ion cluster, the charge on the Mg ion is +1.35, with +0.65 on the six water molecules. The Mg²⁺-

water interaction does thus contain charge transfer via hybridization leading to some dative covalent bonding in addition to the ion-dipole interaction.

Core ionization increases the total charge to +2/+3 for Na/Mg. In the core-ionized $Na^{2+}[H_2O]_6$ ion cluster, the Na ion has charge +1.48 instead of +2, implying +0.52 on the waters. In contrast to the ground state, there is now a charge transfer from the water to the ion, amounting to 0.52 of an electron. For the core-ionized $Mg^{3+}[H_2O]_6$ ion, the +3 total charge is even more distributed, with +1.75 on Mg and +1.25 on the water.

The radiative core-hole decay does not change the total charge, but the charge distribution is entirely different after local $K_\alpha$ decay and IRD, as shown in Fig. 3. For Na and Mg, $K_\alpha$ decay increases the charge of the metal ion and decreases the charge on water, compared to the intermediate core-hole state with equal total charge. In contrast, IRD instead decreases the charge of the metal ion and increases the charge on water. The final state after $K_\alpha$ decay has the majority of the charge on the metal ion, but after IRD, the majority of the charge is instead on the water. This clearly shows that the net effect of the IRD process is

that the charge from the water fills the core hole on the metal ion. In the present context, the charge transfer upon core ionization and core-hole decay can be regarded as instantaneous, but the ultrafast dynamics of this may be possible to probe in the not-too-distant future, using, e.g., attosecond X-ray pulses from free-electron lasers.

We can take the discussion of the IRD process further by analyzing what mechanism provides the intensity of the IRD transitions. The spectral intensity of transitions from the different hybrid orbitals $\psi_\mu$, for a given $\mu$, to the M 1s hole depends on the set of orbital coefficients $C_\mu^\nu$ and the dipole transition moment operator $\mathbf{t}$ projected on the atomic orbital basis set $|\chi\rangle$. The intensity is then proportional to $|\mathbf{T}_\mu|^2 = |\sum_\nu \mathbf{t}_{M1s,\nu} C_\mu^\nu|^2$.

The IRD bands are due to decay from three different hybrid orbitals $\psi_\mu$, which consist mainly of $1b_2$, $3a_1$, and $1b_1$ water outer-valence molecular orbitals, respectively. Of the three IRD lines, the one from $3a_1$ has >10 times higher intensity than the ones from $1b_2$ and $1b_1$ in the idealized geometries studied here, see Supplementary Table 2, and we will consequently focus our discussion on this dominant $3a_1$ line.

In Table 1, we present the main atomic orbital $C_\mu^\nu$ contributions to the hybrid orbitals $\psi_\mu$, the symmetry-allowed Cartesian component of $\mathbf{t}_{M1s,\nu}$—simply denoted by $t_\nu$ hereafter—as well as their product $t_\nu C_\mu^\nu$. Note that in the present structural model, only ungerade hybrid orbitals contribute to the spectra for symmetry reasons. Similarly, the M ns contributions to the hybrid orbitals also do not contribute as $\mathbf{t}_{M1s,Mns} = 0$ due to dipole selection rules. Table 1 forms a basis to discuss the mechanisms behind the different radiative transitions filling the M 1s core hole. Different hybrid orbitals $\psi_\mu$ have different compositions of O 2p and M np contributions, described by the $C_\mu^\nu$ orbital coefficients.

The local $K_\alpha$ emission intensity is due to decay from a hybrid orbital for which the by far largest orbital coefficient is for M 2p, making this an essentially atomic orbital centered on the metal ion. For both Na and Mg, the corresponding transition moment $t_{M2p}$ is 5–10 times higher than any of the other transition moments, resulting in a very high intensity of the $K_\alpha$ line.

In contrast to the $K_\alpha$ case, the $3a_1$ IRD line is due to decay from a hybrid orbital which is strongly dominated by O 2p, with much smaller M np contributions. The final $3a_1^{-1}$ state after the IRD thus primarily consists of a hole delocalized on the water molecules around the metal ion.

As discussed above, the spectral intensity of decay from $\psi_\mu$ is given by $|\mathbf{T}_\mu|^2 = |\sum_\nu \mathbf{t}_{M1s,\nu} C_\mu^\nu|^2$. As seen in Table 1, for $3a_1$ of both Na and Mg, the by far largest $\mathbf{t}_{M1s,\nu} C_\mu^\nu$ term in the sum is due to M np contributions, which are roughly two orders of magnitude larger than

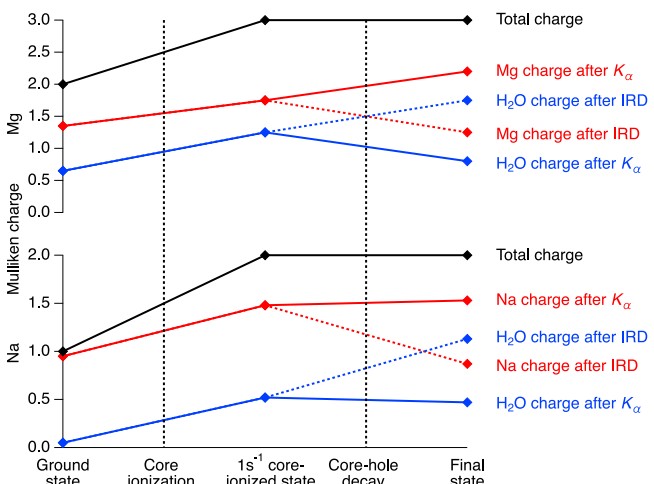

**Fig. 3 | Mulliken charges in the ground, intermediate, and final states for $Na^+[H_2O]_6$ (lower panel) and $Mg^{2+}[H_2O]_6$ (upper panel) for the radiative decay of the core hole.** The total charge (black) is decomposed into charge on the metal ion (red), and on the water molecules (blue). For both metal ion and water, the final-state charge after both local $K_\alpha$ decay and non-local IRD is shown. The connecting lines are only guides for the eye.

**Table 1 | The main contributions to the spectral intensity for $K_\alpha$ and $3a_1$ IRD for the symmetric $M^q[H_2O]_6$ model, with $M^q$ = Na$^+$ and Mg$^{2+}$, in atomic units**

| Metal atom | Na | | | | | Mg | | | | |
|---|---|---|---|---|---|---|---|---|---|---|
| Transition | | $K_\alpha$ | | IRD $3a_1$ | | | $K_\alpha$ | | IRD $3a_1$ | |
| | $t_\nu$ | $C_{M2p}^\nu$ | $t_\nu C_{M2p}^\nu$ | $C_{3a_1}^\nu$ | $t_\nu C_{3a_1}^\nu$ | $t_\nu$ | $C_{M2p}^\nu$ | $t_\nu C_{M2p}^\nu$ | $C_{3a_1}^\nu$ | $t_\nu C_{3a_1}^\nu$ |
| Atomic orbital $\nu$ | $\times 10^{-3}$ | | $\times 10^{-3}$ | | $\times 10^{-3}$ | $\times 10^{-3}$ | | $\times 10^{-3}$ | | $\times 10^{-3}$ |
| M 2p | 49.11 | 0.97 | 47.6 | 0.08 | 3.93 | 47.05 | 0.99 | 46.6 | 0.03 | 1.41 |
| M 3p | −4.53 | 0.01 | −0.045 | −0.08 | 0.362 | −6.78 | −0.01 | 0.068 | −0.35 | 2.37 |
| M 4p | 0.45 | −0.02 | −0.009 | 0.11 | 0.049 | 0.93 | 0.01 | 0.009 | 0.18 | 0.167 |
| O 2p | 0.04 | 0.00 | 0.002 | 0.81 | 0.032 | 0.06 | 0.00 | 0.047 | 0.79 | 0.047 |
| $|\mathbf{T}_\mu|^2 \times 10^{-6}$ | | | 2405.99 | | 17.39 | | | 2216.23 | | 14.74 |

The considered value of the M-O distance $R$ was 2.3 Å for Na and 2.1 Å for Mg. In the columns, $t_\nu$ denotes the respective symmetry-allowed Cartesian component of the atomic dipole transition moments $\mathbf{t}_{M1s,\nu}$ (a. u.) and $C_\mu^\nu$ denotes the orbital coefficient of the main atomic orbitals $\nu$ in the LCAO description of the hybrid orbitals $\psi_\mu$. In the bottom row, $|\mathbf{T}_\mu|^2 = |\sum_\nu \mathbf{t}_\nu C_\mu^\nu|^2$ is proportional to the total spectral intensity of the $K_\alpha$ and $3a_1$ IRD transitions. Note that $|\mathbf{T}_\mu|^2$ also includes minor contributions from M np orbitals with n larger than 4. Data are from molecular orbitals optimized for the ground state by HF.

the ones related to O nl. The different M np contributions differ in the sense that M 2p is fully occupied already in the free ions, while M 3p and M 4p are partially occupied in the solvated ion due to hybridization with the water molecular orbitals. For Na, the $\mathbf{t}_{M1s,\nu} C_\mu^\nu$ term associated with Na 2p is ten times higher than any other, whereas for Mg the $\mathbf{t}_{M1s,\nu} C_\mu^\nu$ term associated with Mg 3p is somewhat higher than for Mg 2p. This is consistent with the above-discussed larger ground-state charge transfer from water to the metal ion for $Mg^{2+}$ than for $Na^+$, resulting from the increased ion-water hybridization with decreasing ion-water distance. For Na, the intensity thus mainly comes from Na 2p–O 2p hybridization, while for Mg also ground-state charge transfer into Mg 3p is important. The intensity in IRD is therefore mainly due to transitions from the metal atomic orbital components of the hybrid orbitals. The small $C_\mu^\nu$ metal orbital coefficients of the $\psi_\mu$ hybrid orbitals are compensated by the atomic transition moments $t_\nu$ being much larger for the metal orbitals than for the water orbitals.

Based on the theory presented above, our interpretation is that the electron filling the core hole originates from the water molecules in the first solvation shell, and the IRD signal strength is attributed to hybridization with the metal-ion orbitals. The hybrid orbitals have relatively small metal character, implying that the electron is to a great extent localized on the water molecules. However, the associated high dipole transition moment of the metal component of the hybrid orbital results in a much higher contribution to the intensity than from the water component, which instead provides a very low dipole transition moment. At the same time, the final-state hole is mainly delocalized on the nearest water molecules, meaning that the electron filling the $M^{q+1}$ 1s core hole effectively mainly comes from the solvation shell water, and its energy reflects such delocalization. In spite of this non-local character of IRD, the strong contribution to the intensity from the small metal character of the hybrid orbitals makes the one-center approximation a good approximation, as elaborated further in Supplementary Note 2.3. Moreover, by varying the M−O distance, as discussed in Supplementary Note 2.5, the IRD efficiency is seen to decrease rapidly with increasing distance, due to a decrease of the metal-water orbital mixing. This means that IRD, to a good approximation, only involves the first solvation shell.

The above discussion is based on calculations using an idealized, highly symmetric structure, which is useful to capture the essentials of the IRD mechanism. For a similar analysis using a more realistic structure model suggested by molecular dynamics (MD), see Supplementary Note 2.4.

## IRD and the electronic structure

As we will show, IRD opens new possibilities for chemically selective probing the electronic structure of the first solvation shell around a solvated ion. Starting with bulk liquid water, the electronic structure has been studied by valence-band photoelectron spectroscopy (PES), providing the $H_2O^+$ val$^{-1}$ energies, see, e.g., Ref. 34. The valence-band photoelectron spectrum of liquid water, shown in Fig. 4, consists of three features. These are associated to ionization of the three outermost valence orbitals of the water molecule, 1b$_1$, 3a$_1$, and 1b$_2$, modified by overlap and screening in the liquid environment[35]. As a simplistic first approximation, the solvation-shell $H_2O^+_{solv}$ val$^{-1}$ binding energies would be expected to increase around a positively charged ion, but we should also consider that the geometrical arrangement of the close to six water molecules surrounding a cation is quite different from that of the tetrahedral-like structure of the close to four water molecules surrounding a typical water molecule in bulk liquid water. The water-water orbital interactions can thus be expected to be different in the solvation shell, and in addition there may be ion-water orbital interactions, including polarization of the orbitals by the electric field of the ion. We can obtain the corresponding one-hole energies for the solvation-shell water, $E(H_2O^+_{solv}$ val$^{-1})$, from the IRD spectra as the difference between the M 1s binding energy of the metal ion, $E(M$ 1s$^{-1})$,

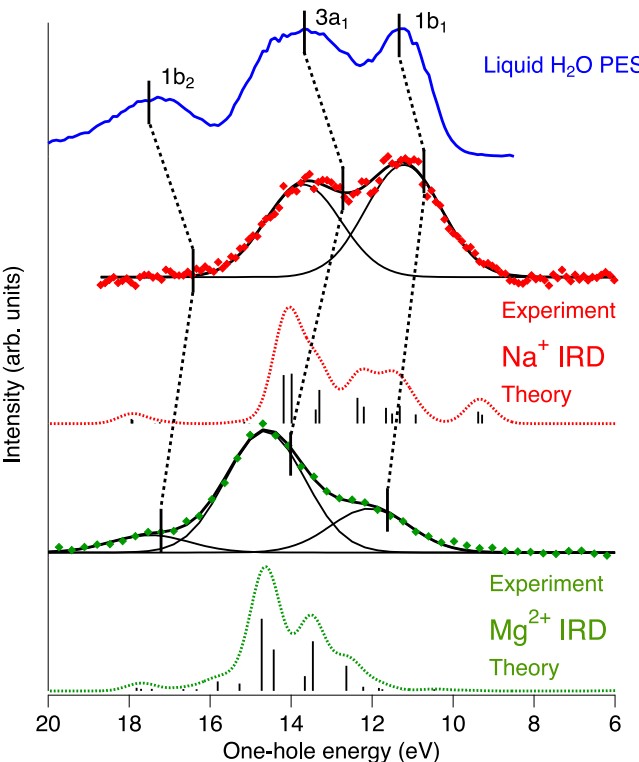

**Fig. 4 | Comparison of one-hole spectra of H₂O, Na⁺, and Mg²⁺.** From the top, liquid-water PES spectrum recorded using a photon energy of 532 eV, i.e., under relatively bulk-sensitive conditions[51] (blue), Na IRD spectrum (red dots, experiment, and red dashed, theory), Mg IRD spectrum (green dots, experiment, and green dashed, theory). Experimental and theoretical IRD spectra are enlargements of the data shown in Fig. 2. Solid black curves indicate a fitting of the experimental data using a minimal number of Gaussian curves. Original theoretical data for Na and Mg are represented by black bars, and the dashed curves for their convolution with a Gaussian function (FWHM = 0.8 eV) to facilitate the comparison with the experimental spectra. Vertical solid lines indicate calculated binding energies from Ref. 14.

and the photon energy of the IRD transitions, hν(IRD); $E(H_2O^+_{solv}$ val$^{-1})$ = $E(M$ 1s$^{-1})$ − hν(IRD).

Turning to the Na IRD spectrum, shown in Fig. 4, it is seen to exhibit two peaks with energies very close to the two most intense peaks of bulk water. As for bulk water, we provisionally interpret these as corresponding to holes in the 3a$_1$ and 1b$_1$ orbitals in the first solvation shell. For Mg, there are three peaks, of which the two stronger ones exhibit clear shifts to higher one-hole energies relative to bulk water.

Figure 4 also shows calculated IRD spectra, obtained by HF calculations for $Na^+[H_2O]_6$ and $Mg^{2+}[H_2O]_6$ in a representative cluster geometry derived from MD simulations, see structure in the middle of Supplementary Fig. 3. For both ions, the calculations and the experiments agree quite well in terms of energies. For the spectral shape, there is also a good agreement for Mg with 3a$_1$ dominating over 1b$_1$ and 1b$_2$. For Na, the calculated spectrum also peaks in the 3a$_1$ range, whereas for the experimental spectrum 1b$_1$ is even stronger than 3a$_1$. We tentatively ascribe this discrepancy as due to orientational disorder of the water molecules in the solvation shell, as discussed in Supplementary Note 2.6.

It should be noted, however, that the calculations show that the experimentally measured peaks cannot uniquely be assigned to holes in the 1b$_1$, 3a$_1$, and 1b$_2$ molecular orbitals of individual solvation-shell molecules. Instead, the final one-hole states are delocalized over two or more solvation-shell molecules. The orbital composition and energy

of the final one-hole states also vary with the different relative orientations of the water molecules. This means that spectral contributions from holes in the three valence molecular orbitals of water overlap. That said, the relative contribution of $1b_1$ holes is highest for the low-energy states, whereas the relative contribution of $3a_1$ holes is highest for states of higher energy. We will, therefore, with the above reservations in mind, continue to denote the IRD spectral features as $1b_1$, $3a_1$, and $1b_2$.

We can also compare the observed $H_2O^+_{solv}$ val$^{-1}$ energies for Na and Mg with calculated energies from Ref. [14], shown as vertical lines in Fig. [4]. Contrary to the simplistic expectation, the calculations predict a small shift towards lower one-hole energies relative to pure water for Na. This was explained as due to the structure-breaking properties of Na, causing a disruption of the water hydrogen-bond network.

For Mg, our results and the calculations from Ref. [14] agree upon a shift towards higher energy relative to pure water. For both ions, the calculated energies from Ref. [14] are somewhat lower than our observed ones. This may at least partly be due to the experimentally obtained $H_2O^+_{solv}$ val$^{-1}$ states being vibrationally excited, whereas the calculated energies are for the vibrational ground states.

The present results based on IRD can also be discussed relative to the O 1s X-ray Emission Spectroscopy (XES) spectra of solvation-shell water obtained by subtracting a suitably scaled spectrum for pure water from a spectrum of a MgCl$_2$ solution[36]. The two approaches probe the same electronic final states, $H_2O^+_{solv}$ val$^{-1}$, but in different ways. The spectra obtained by the subtractive procedure in Ref. [36] contain contributions from solvation shells around both Mg$^{2+}$ and Cl$^-$ ions, whereas IRD probes the solvation shells around the different ions selectively. The spectral shapes are also quite different, as the processes are very different, $M^{q+1}1s^{-1} + H_2O_{solv} \rightarrow M^q + H_2O^+_{solv}$val$^{-1} + h\nu$ for the present IRD spectra, versus $H_2O^+_{solv}O\ 1s^{-1} \rightarrow H_2O^+_{solv}$val$^{-1} + h\nu$ for the differential O 1s XES spectra of Ref. [36]. This results in different relative intensities of $H_2O^+_{solv}$val$^{-1}$ electronic states, as well as different populations of vibrational states. The spectra obtained by the subtractive procedure are also influenced by the nuclear dynamics during the O 1s core-hole lifetime, a complication not affecting the IRD spectra.

IRD appears to be a general phenomena for ions in liquid surrounding, in addition to these ions with neon-like electronic configuration, we have also observed IRD for transition metal ions (Cu$^{2+}$) and anions (F$^-$) in aqueous surrounding, see Supplementary Note 1.2.

We conclude that IRD can be used to selectively probe the $H_2O^+_{solv}$val$^{-1}$ energies of the first solvation shell around ions, revealing how the electronic structure differs from bulk water. This would also include the presence of other species, e.g., contact ion pairs, see Supplementary Note 2.7.

## Discussion

Combining experiments and calculations, we have demonstrated the existence of IRD for Na$^+$ and Mg$^{2+}$ ions in aqueous solution. In IRD, a 1s core hole on Na or Mg is filled by an electron from a neighboring species, and an X-ray photon is emitted. IRD is thus a non-local decay process, which in some sense can be considered a radiative analog of ICD.

The transitions observed in the IRD spectra are interpreted as getting intensity from hybridization between valence orbitals on water molecules and the occupied orbitals on the metal cation. The associated high atomic transition moments result in a much higher contribution to the intensity than from the dominant oxygen character. At the same time, the final-state hole is delocalized on the nearest water molecules, meaning that the electron filling the metal 1s core hole effectively comes from water. For the observed IRD lines, the intensity comes from the relatively small electron density on the metal ion, while the energy position derives from the electron density that is largely delocalized on the water molecules.

These results for hydrated Na$^+$ and Mg$^{2+}$ ions demonstrate the potential of IRD as a more generally applicable method where it is possible to use the solvated ions to selectively probe the solvation shell around each species, thereby opening new possibilities to understand how the solvation shell affects the chemical properties of solutes.

## Methods
### Experimental

The measurements for Na$^+$ and Mg$^{2+}$ in water were performed using a cylindrical liquid jet at the VERITAS beamline at the MAX IV synchrotron radiation facility in Lund, Sweden[37]. The VERITAS beamline comprises an elliptically polarizing undulator and a collimated plane-grating monochromator with an ellipsoidal refocusing mirror. The beamline is equipped with a large constant-line-spacing grating Rowland spectrometer with a collimating mirror in the non-dispersive direction and a cylindrical grating with 1400 grooves/mm and 67 m radius[38]. Measurements were done with linear horizontal polarization of the incident radiation.

The samples, 2 M NaCl and 2 M MgCl$_2$ solution, were prepared by dissolving commercially purchased NaCl and MgCl$_2$ (Sigma-Aldrich) with purity of >98% in MilliQ water with a resistivity of 18.2 M$\Omega \times$ cm. The sample was introduced via a vertically mounted liquid jet shooting into a cold trap cooled by liquid nitrogen. The liquid jet was surrounded by a cylindrical differential pumping stage with holes for incident and outgoing photons, allowing the sample to be intersected by the X-rays from the beamline in front of the soft X-ray spectrometer.

Ionization of the 1s level of the Na$^+$ and Mg$^{2+}$ ions was done using photon energies of 1083 and 1318 eV, respectively, i.e., 6 and 8 eV above the respective M 1s thresholds. In both cases, these photon energies are well below the thresholds for the formation of higher-energy states, like M 1s$^{-1}$2p$^{-1}$3p or M 1s$^{-1}$2p$^{-1}$, ensuring that M 1s$^{-1}$ is the only state from which the decay occurs. Emitted photons were detected in the plane of the polarization of the incident radiation at an angle of 90° with respect to the propagation axis. Calibration of the emitted photon energies was done using the energy of the K$_\alpha$ transition, M 1s$^{-1} \rightarrow$ M 2p$^{-1} + h\nu$, to set the absolute energy at one point of the detector, and a set of elastic peaks to determine the dispersion over the detector. The energy of the K$_\alpha$ transition was determined from the binding energy difference between the M 1s$^{-1}$ and M 2p$^{-1}$ states of solvated Na$^+$ and Mg$^{2+}$ ions, respectively, obtained by electron spectroscopy[14]. For Na$^+$ and Mg$^{2+}$, we thus obtain h$\nu$(K$_\alpha$) as 1041.3 eV (1076.7−35.4) and 1254.3 eV (1309.9−55.6), respectively[14]. This procedure relies on the energy difference between the elastic peaks and not their absolute energy, thereby becoming independent of the absolute monochromator energy scale. An upper limit of the spectrometer resolution is the total observed width of the K$_\alpha$ peaks. For both the Na$^+$ and Mg$^{2+}$ edges, this is less than 0.6 eV, which is significantly smaller than the width of the fitted components of the IRD signals (-2.2 eV). The X-ray emission spectra of Na and Mg were measured in the third and second orders of diffraction, respectively.

A pilot study for Mg$^{2+}$ in water was also performed at the P04 beamline of the synchrotron radiation facility PETRA III (DESY, Hamburg, Germany), see Supplementary Note 1.1. The result from P04 is consistent with the results from VERITAS, on which the analysis in this paper is based.

The valence-band photoelectron spectrum of water was recorded at the FlexPES beamline of MAX IV, Lund, Sweden[39], using a photon energy of 532 eV. The photon energy resolution was -0.23 eV, and the spectrometer resolution -0.25 eV, resulting in a total resolution of -0.35 eV. To reduce spectral contributions from gas-phase water, a bias voltage of −50 V was applied to the jet when recording the spectrum.

Preliminary experiments for Mg$^{2+}$ in water were carried out at the P04 beamline of the synchrotron radiation facility PETRA III, DESY, Hamburg[40], using a setup combining a cylindrical liquid jet and a soft X-ray spectrometer[41]. The sample, 2 M MgCl$_2$ solution, was prepared by

dissolving commercially purchased MgCl$_2$ (Sigma-Aldrich with purity of > 98%) in MilliQ (18.2 M$\Omega$ × cm) water. The sample was introduced via a vertically mounted liquid jet shooting into a liquid-nitrogen-cooled cold trap. The liquid jet was intersected by the circularly polarized X-rays from the beamline in front of the soft X-ray spectrometer. The spectrometer was a modified grazing-incidence Rowland-circle spherical grating spectrometer (GRACE) equipped with a 1200 l/mm grating[41]. The X-ray emission spectra were measured in fourth order. The beamline and the X-ray detector were protected from the relatively high pressure in the measurement chamber during liquid-jet operation by a differential pumping stage and a filter between the grating and the detector, respectively. Due to the presence of the filter, the internal motions in the spectrometer were impeded. Due to the compromised focusing due to the impeded motions, the spectral resolution was reduced to ~6 eV, which limits the ability to resolve fine substructure in the IRD region.

## Computational

Theoretical investigations were conducted to simulate the experimental spectra and to understand the mechanisms behind the IRD effect; a number of model structures and computational methods have been employed to this end. Two kinds of cluster models were considered: small idealized systems and ion/water clusters inferred by MD simulations of aqueous solutions of NaCl and MgCl$_2$.

By a reasonable assumption about the interaction between the positive charge of the ions and the dipole of the water molecules, we firstly considered idealized structures in which a few water molecules are oriented with the dipole vector along the M$^q$–O axis ($R$), in a planar geometry for M$^q$[H$_2$O]$_4$, and in an octahedral geometry for M$^q$[H$_2$O]$_6$. Using such small systems, as M$^q$[H$_2$O]$_n$, with $n$ = 1, 4, 6, the role of ion-water distance, symmetry, and number of molecules surrounding the cation was studied. Transition energies and radiative transition intensities were computed within the RASSCF level of theory, as implemented in the OpenMolcas package[42], together with the aug-cc-pVTZ basis set and using the HF wavefunction as reference (HF/RAS). For the different M$^q$[H$_2$O]$_n$ cases, the RAS subspaces were built with the M$^q$ 1s orbital in RAS1, the three M$^q$ 2p orbitals in RAS2, and the 4$n$ water molecular orbitals corresponding to 2a$_1$, 1b$_2$, 3a$_1$, and 1b$_1$ of the $n$ water molecules in RAS3, in such a way that the ionized system had a total of 2(4 + 4$n$)−1 active electrons in 4 + 4$n$ orbitals. Relativistic effects based on the Douglas-Kroll decomposition of the electronic Hamiltonian and spin−orbit couplings were included in both calculations.

To go beyond the idealized symmetrical structures and therefore determine a more realistic structural model of Na$^+$ or Mg$^{2+}$ ions solvated in water, MD calculations were carried out with the GROMACS code version 5.1.4[43] on two samples, both contained in a cubic box with a side length of 50 Å and periodicity conditions in the three dimensions. The sample, which was intended to simulate a situation of extreme dilution, contained a single M$^q$ ion and the appropriate number of Cl$^-$ counter ions, and 4166 water molecules, while in the second case, the number of M$^q$ ions was brought to 150 and the counter ions increased correspondingly, to replicate the concentration of the solution (2 M) used experimentally.

For the M$^q$ and Cl$^-$ ions, the force field OPLS/AA[44] has been used, while the water molecule has been simulated with the TIP4P model[45]. Starting from a random distribution of the molecules in the solution, a standard minimization-heating-equilibration protocol was adopted, followed by an extended production run of 10 ns with a time step of 1 fs. Apart from the heating-equilibration steps, the canonical ensemble (NVT) dynamics were conducted at T = 300 K fixed with a velocity-rescaling thermostat. The analysis of the MD simulation carried out on the first sample confirms the result of a previous accurate quantum MD calculation[46], i.e., that, at high dilution, the M$^q$ ion is surrounded by six water molecules with high coordination. In contrast, the second hydration shell is less ordered and more sensitive to rearrangements at

room temperature. From the analysis of the MD trajectories of the larger samples, using the cluster tool of GROMACS, the most typical structures of the solvation shells of M$^q$ at the 2 M concentration were identified.

This allowed us to select some specific clusters containing a single M$^q$ ion and a variable number of water molecules for which the XES spectrum generated by the ionization of the M$^q$ core shell was calculated. For these larger model systems, we adopted the independent particle approximation, where each state is described by a single determinant, namely the less computationally expensive HF and DFT methods, to investigate the dependence of the IRD bands on the number of solvation shells, the orientation of the water molecules closer to the ions, and the presence of ion paring. This was accomplished in the Kohn-Sham DFT ground-state approximation using the StoBe-deMon code[47], the Perdew-86 density functional, and a TZVP basis set[48]. Such a method can be easily applied to large systems, but is affected by the well-known self-interaction error for the core electron.

In the independent-particle approximation, i.e., neglecting electron correlation, the XES spectrum is obtained by HF calculations for the ground state and the core-hole state. For the latter, the important electronic relaxation is described by optimizing the electronic structure under the constraint of the single occupancy of the core orbital. The final states of the emission process are instead approximated as Koopmans' states, i.e., by removing an electron from one of the frozen valence orbitals of the ground state, thus ignoring the negligible electronic relaxation. The intensity of the radiative process is then calculated, in the long-wavelength approximation, from the transition dipole moment between the initial state and the final states projected onto sets of non-orthogonal orbitals. The HF calculations have been performed by using the DALTON code[49] with the Ahlrichs-VTZ basis set, while the transition moment has been computed by an homemade code. The restricted open-shell HF approximation, fully including the electron relaxation around the initial core hole, can be considered, despite missing any electron correlation effect, to provide a more reliable simulation of the XES spectra. However, the present calculations found that both DFT methods, with an appropriate energy shift, and HF methods predict quite similar XES spectra.

Comparison of simulated and observed experimental spectra rationalized the mechanisms behind the IRD process. The validity of the one-center approximation for IRD was also considered, see Supplementary Note 2.3, in comparison with full calculations.

## Data availability

The theoretical and processed experimental data needed to re-generate the figures in this study have been deposited at Zenodo Ref. 50. The raw experimental data is available upon request from the authors. The input files and molecular geometries needed to run the theoretical modeling is also available in Ref. 50. Source data are provided with this paper.

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

## Acknowledgements

We acknowledge MAX IV Laboratory for time on the VERITAS beamline under proposal 2022-1207. Research conducted at MAX IV, a Swedish national user facility, is supported by the Swedish Research Council under contract 2018-07152, the Swedish Governmental Agency for Innovation Systems under contract 2018-04969, and FORMAS under contract 2019-02496. We acknowledge DESY (Hamburg, Germany), a member of the Helmholtz Association HGF, for the provision of experimental facilities. Parts of this research were carried out at PETRA III and would like to thank Moritz Hoesch and his team for assistance in using beamline P04. Beamtime was allocated for proposal I-20211465 EC. O.B., J.-E.R., and H.Å. acknowledge funding from the Swedish Research Council (VR) through the projects VR 2023-04346, VR 2021-04017, and VR 2022-03405, respectively, as well as the Swedish Foundation for International Cooperation in Research and Higher Education (STINT) through project 202100-2932. The authors also thank the Swedish National Infrastructure for Computing (NAISS 2022/3-34) at the National Supercomputer Center of Linköping University (Sweden) partially funded by the Swedish Research Council through grant agreement no. 2018-05973. M.H. acknowledges funding by the German Federal Ministry of Education and Research under grant number 13K22XXB DYLIXUT. D.B. and A.H. acknowledge support by the German Federal Ministry of Education and Research (BMBF) through project 05K22RK1-TRANSALP. F.T. acknowledges funding by the Deutsche Forschungsgemeinschaft (DFG, German Research Foundation) - Project No. 509471550, Emmy Noether Program. A.N.B. acknowledges funding from FAPESP-Brazil - Proj. 2017/11986-5, 2024/00998-6 and FAEPEX-UNICAMP 3358/24, 2684/25. A.N.B. and L.M.C. acknowledge FAPESP 2021/06527-7 and 2020/04822-9. R.M. acknowledges funding from the Federal District Research Foundation (FAPDF) for the 410/2022 project.

## Author contributions

J.S., J.-E.R., and O.B. conceived the experiment. J.S., M.A., D.B., F.T., I.I., D.V., J.P., J.-E.R., M.H., Z.Y., J.N., and O.B. performed the initial measurements at P04. J.S., V.E., O.B., A.N.d.B., M.A., R.M., T.T., C.S., Y.-P.C., and A.G. performed the main experiment at VERITAS. G.Ö. provided the photoemission data. J.S. did the data analysis. L.M.C., V.C., and H.Å. performed the theoretical modeling. J.S., O.B., L.M.C, V.C., and H.Å. wrote the manuscript. J.S., M.A., J.-E.R., A.N.d.B., Z.Y., A.H., J.N., L.M.C., V.C., H.Å., and O.B. discussed the interpretation and provided input on the manuscript.

## Funding

## Competing interests

The authors declare no competing interests.
