## [Transparent Peer Review file · Nature Communications]

Non-local X-ray intermolecular radiative decay probes solvation shell of ions in water

Corresponding Author: Dr Johan Söderström

Version 0:

Reviewer comments:

Reviewer #1

(Remarks to the Author)

The article by Söderström et al. reports the first observation of intermolecular radiative decay in aqueous Na⁺ and Mg²⁺ ions, where a 1s core hole on the metal ions is refilled by an electron from nearby water molecules in the first solvation shell. The authors provide a theoretical analysis of this mechanism and claim that this newly identified effect offers valuable insights into the electronic structure of solvated molecules. While I acknowledge the pioneering nature of this observation and the initial analysis, I have concerns regarding the method's transferability to other systems, the depth of the analysis, and the article's structure and style. The discussion in the main text is overly simplistic, with key details relegated to the supplement, making the main text uninformative when read on its own. Combined with certain technical issues, I find the arguments insufficiently convincing to justify publication in Nature Communications at least in its current form. My detailed comments follow below.

The authors present the observed effect as broadly applicable. Indeed, it appears better suited than interatomic Coulombic decay (ICD) for probing the electronic structure of solution bulk. However, its general applicability is questionable. For the phenomenon to occur, the occupied frontier orbitals of the solvent must lie significantly higher in energy than the metal's occupied 2p orbitals to isolate non-local signatures from local metal decay. This naturally occurs in the studied systems due to the highly stable Ne-like configuration of the metal ions. One might expect the effect to be relevant in cases of high ionicity and positive charge, but is it transferable to other systems? In more covalently bound environments, these orbitals approach near-degeneracy, preventing clear signal isolation. For instance, in transition metal ions probed at the L-edge, the local d orbitals are nearly degenerate with the solvent orbitals, making separation difficult. Additionally, in anionic systems, the signal could appear below the local decay. Does this limit the method's applicability? Can it effectively probe solvation in neutral or merely polar (non-ionic) solutes? These questions require clarification, and the claims should be adjusted accordingly. As it stands, the method does not unequivocally "offer a chemically selective way to probe the electronic structure," contrary to what is stated in the article (see my next comment).

The authors claim that the method is well suited for probing electronic structure. However, they experience difficulties in drawing clear conclusions due to overlapping spectral bands and in explaining energy shifts and intensity variations. If such complications arise in these relatively simple systems, how can the method be reliably applied to more complex cases? A rigorous analysis, such as for an octahedral solvation shell, would involve distinguishing different groups of water orbitals. Based on symmetry considerations, the 2a₁, 1b₁, 3a₁, and 1b₂ orbitals should form distinct bands, split into singly, doubly, and triply degenerate groups according to octahedral symmetry representations. Hydrogen-bonding lone pairs (3a₁ and 1b₁) would likely show the most significant energy splitting and potential overlap. The conclusions on electronic structure should also address nuclear dynamics-related factors, such as the statistical distribution of first-shell water molecules and contributions from contact pairs. While theoretical studies on these aspects are extensive, the main text merely mentions them, relegating details entirely to the supplement. If such analysis proves difficult even for these simple systems, the claim of providing "an insider's view of the solvation shell" should be significantly tempered. Moreover, as the study examines electronically relaxed species post-ionization, it remains unclear how this relates to the original solvated state.

Regarding the computational setup, the discussion of Hartree-Fock (HF), RASSCF, and DFT calculations is somewhat disorganized, making it unclear which method was used for which aspect of the study. From the explanations, it appears that what the authors refer to as "HF" is actually a configuration interaction singles (CIS) method with orbital optimization enabled via RAS subspaces. Otherwise, how could excited single-hole states be computed with HF? Additionally, the

rationale for using DFT to compute intensities is unclear when OpenMolcas could handle them directly, even incorporating perturbation theory to improve correlation effects. Concerning correlation, the active space could be expanded to include unoccupied orbitals, such as the 3s of Na or Mg and two antibonding water orbitals, to capture additional correlation effects. Even more crucially, including the metal 3p orbitals would provide a stronger theoretical foundation for mechanism 2 on page 6 of the supplement. While correlated treatments (both RASSCF and RASPT2) may only subtly alter intensities by mixing configurations with electrons in the metal's unoccupied 3p orbitals, such adjustments could be decisive given the already weak transition intensities. The authors also briefly mention vibronic effects, but these should significantly impact both intensities and energy shifts due to substantial structural differences between $M^{(q+1)}-H_2O$ and $M^q-H_2O^+$ species.

The analysis of Mulliken charges as an attempt to explain intensity borrowing is misleading. Transition intensities depend on dipole matrix elements, not charge distributions. The correct explanation appears only in the supplement, though it focuses on orbitals rather than many-electron states. Additionally, the mention of the one-center approximation in the context of a non-local effect is confusing. While the supplement clarifies the authors' intended meaning, the main text's brief reference is insufficient and could be misinterpreted. Furthermore, the statement that "the intensity and the electron come from different sources" is poorly worded and misleading—intensity and electron transfer are distinct concepts that should not be conflated.

Overall, the main text contains redundant and overly simplified explanations (e.g., the section "The IRD mechanism," particularly the paragraph following line 204), making the intended audience unclear. Meanwhile, most of the substantive discussion is confined to the supplement. Ideally, this should be reversed: the main text should provide the core scientific discussion, with supplementary material used for additional details. As it stands, the main text simply lists what was done, whereas the supplement contains the actual scientific discourse.

Minor:

- In the introduction, the paragraph beginning "The electronic configuration of the free metal..." is too abrupt. Why were these two ions chosen? Why is the charge $q+1$? Basic explanations from elsewhere in the article could be relocated here.
- Lines 118–119 discuss the 2p orbitals of free ions, yet the conclusion pertains to solvated ions.
- The caption of Figure 2 mentions a shift in computed spectra but does not specify the value.
- In line 217, orbitals are referred to as "metal-centered" despite being described as having "small metal character."
- In line 242, emission energies are linked to electron binding energies; however, ionizing and non-ionizing transitions are not equivalent processes.

Reviewer #2

(Remarks to the Author)

This article involves a joint experimental and theoretical investigation of a new process known as intermolecular radiative decay (IRD) occurring for metal (Na⁺/Mg²⁺) ions in water. The process involves initial ionization of the 1s core-hole state of the metal ion, which induces relaxation of a valence electron residing in the solvation shell and emission of a photon. The hypothesis is that this process can be used as a useful probe of the local solvation shell.

Unfortunately, there are some key limitations both in the quality of the data and in the structure of the paper, which makes it difficult to recommend the paper for publication in Nature Communications as outlined below:

1. The key experimental findings of the manuscript are associated with the results presented in figure 2 and 4 associated with the spectra assigned to IRD. The author's claim that these spectra can be used as a useful tool to examine the electronic structure of the solvation shell local to the metal ions in comparison to bulk water. However, the noise observed in the spectrum, especially for Na⁺ spectrum (associated with the oscillations) seemingly makes it impossible to distinguish if any features change from the bulk structure. Even the Mg spectrum appears to have very sharp features, for example the very pointed peak for the 3a₁ part of the spectrum, which would seem to indicate that only a few points are used to plot the spectrum. To draw any conclusions, it would be necessary to plot some form of error bars on the spectrum. It would also be beneficial to plot a difference between the bulk and the IRD spectrum as well.
2. The authors do not hypothesize nor discuss what other processes may be contributing to the observed signal. For example, is it possible to have resonant hole transfer to the solvation shell to generate an excited ionized state of the neighboring solvation shell, which can then subsequently decay?
3. Given that it is known, and the authors even state, that ultrafast non-radiative processes such as Auger, ICD, and ETMD occur in these systems, it is surprising that any radiative process is observed. Are the authors able to make any comment to the relative timescale of the IRD process in comparison to the other known competing processes? This would help solidify why IRD is even feasible in these systems.
4. Figure 3 presents the Mulliken charges for the various states for K α decay vs IRD. The authors make conclusions about the nature of charge-transfer for IRD based on these results. However, it seems like that this is the only possible conclusion that could have been drawn given that the authors are specifically choosing what the initial and final states are along the pathway. i.e. they are choosing that the IRD final state specifically corresponds to one involving the electron density on water. Now maybe this isn't the case, but the calculation details are so limited in scope (see point 5) that it's not possible to tell if this is a trivial result.
5. The presentation of the calculation details is both disorganized, appearing separately in the methods section as well as

various sections of the SI, and without sufficient detail to understand how the calculations are being performed. The paper needs to be reorganized such that its clear what calculations are being exactly performed and what is the connection between all the seemingly disparate calculations. For example:

a. The only comments to the type of calculations used in the analysis in the results and discussion section seems to be the phrase "X-ray emission spectra for Na⁺ and Mg²⁺ ions within a cluster of six water molecules have been calculated in the Hartree-Fock (HF) approximation" on page 4 and "Figure 4 also shows calculated IRD spectra, obtained by HF calculations for ..." on page 8. However, the methods section provides details on RASSCF and DFT calculations, which does not match the comments about HF in the results and discussion section.

b. The authors make a statement that RASSCF was used to calculate radiative decay rates. However, there was no discussion or comment made in the main text about the decay rates.

c. There is no clear discussion about how the emission energies are obtained. If they're at the RASSCF level, how were the specific ionized states chosen? If they're obtained specifically from a pure HF calculation, is this done by simply looking at the MO energies of a ground-state HF calculation? Significantly more detail is necessary because as of now it would be impossible to reproduce the results.

d. The analysis performed in section 2.4.1 seems to involve calculating matrix elements between single molecular orbitals. However, like point c, no explicit comment has been made as to where the MOs are obtained. Are these MOs obtained from a ground-state HF calculation?

e. Section 2.4.2 makes comments to a pure HF calculation, but it is unclear how these calculations relate to any previous electronic structure calculations as they use a different basis set. There is also no detail provided as to the specifics of these calculations. Does this involve a pure ground-state calculation? This doesn't seem correct as the authors comment that they are somehow targeting "the relaxed M1s-1 state" which would not correspond to a ground-state configuration.

f. Section 2.5 describes the use of an additional basis set that once again has no connection to the previous sets of calculations.

g. Additionally, the authors should include the geometries and, if possible, input files of any electronic structure calculation performed to allow for ease of reproducibility.

A few additional more minor comments:

6. The authors should define what q is on page 3 in the notation M^{q+1} i.e. that q corresponds to the original charge of the cation prior to core ionization.

7. How did the authors choose the representative configuration from the molecular-dynamics trajectory?

8. The following phrase "The HF approximation, fully including the electron relaxation around the tinal core hole..." on page 13 is strangely worded. Do the authors mean to say if some % of exact HF exchange is included in the DFT functional? Electron relaxation is inherently included in any SCF calculation whether it's DFT or HF.

Reviewer #3

(Remarks to the Author)

This article reports on a highly original observation, which identifies X-ray emissions of core-excited aqueous solutions of Na⁺ and Mg⁺⁺ ions lying about the K α emission as a valence solvent to metal core orbital filling via emission of light. To my knowledge, this represents a new type of observable of the solvation shell, which has so far been lacking (or been indirect) in X-ray studies of solvated species. The arguments put forward by the authors and the theory solidly support their interpretation. It would be now interesting to extend such studies to other solutes and eventually, other solvents too, such as alcohols. I strongly support publication without changes.

Reviewer #4

(Remarks to the Author)

This is a well-written and carefully conducted study. The authors support their experimental findings with convincing theoretical analysis, and the results are presented very clearly. My main issue with this paper is its claimed novelty as stated by the authors 'Here, we report experimental results on a novel non-local X-ray emission process, Intermolecular Radiative Decay (IRD).'

The phenomenon of non-local x-ray emission described and named IRD by the authors is not new. Already in 1966, Best (Best, Electronic Structure of the MnO₄⁻, CrO₄²⁻, and VO₄³⁻ Ions from the Metal K X-Ray Spectra, Chem. Phys. 44 3248 (1966)) and in 1975 Jones and Urch (Jones and Urch, Metal-ligand bonding in some vanadium compounds: a study based on X-ray emission data, J. Chem. Soc., Dalton Trans. 1885 (1975)) assigned weak x-ray emission lines (~ 50 eV above the K β main line) observed in transition metal oxides as ligand 2s to metal 1s 'interatomic' or 'crossover' transitions. Best starts the abstract of his paper with 'New measurements of the K x-ray emission spectra from the metal atoms in MnO₄⁻, CrO₄²⁻,

and VO₄²⁻ ions are presented and discussed in terms of dipole transitions from occupied orbitals to the 1a₁, i.e., the metal 1s vacancy.'

A detailed study of the intensity and spectral position of these interatomic emission lines related to the metal-ligand distance has been published in 1999 for a series of Mn oxides (Bergmann et al, Chemical Dependence of Interatomic X-Ray Transition Energies and Intensities - A Study of Mn Kβ¹ and Kβ_{2,5} Spectra, Chemical Physics Letters 302, 119 (1999)). Since then, these transitions have been applied numerous times to identify and characterize ligand atoms (see, e.g. Pollock & DeBeer, Valence-to-core X-ray emission spectroscopy: a sensitive probe of the nature of a bound ligand, Journal of the American Chemical Society 133, 5594 (2015)), led to an important breakthrough in the structure of the FeMoCo cluster in nitrogenase by identifying the central carbon atom (Lancaster et al, X-Ray Emission Spectroscopy Evidences a Central Carbon in the Nitrogenase Iron-Molybdenum Cofactor, Science 334, 974-977 (2011)), and were employed to directly measure the oxygen ligands in the Mn cluster of the photosystem II protein in a water solution (Pushkar et al, Direct Detection of Oxygen Ligation to the Mn₄Ca Cluster of Photosystem II by X-ray Emission Spectroscopy, Angew Chem Int Ed. 49, 800-803 (2010)). Several other studies have been reported as these transitions have become a powerful tool in x-ray spectroscopy.

While in these examples transition metal complexes are studied, the underlying process is identical to what the authors call IRD as described in the submitted manuscript, namely a 1s core hole of the element that is probed is filled by an electron from a neighboring atom or molecule (in the submitted case the water around the ion, in previous cases ligand atoms of molecules). From my understanding, the treatment for calculating the transition energies and strengths is also the same as in those previously published studies. I therefore do not agree with the claim by the authors in their conclusions 'Combining experiments and calculations, we have demonstrated the existence of a novel core-hole decay process, Intermolecular Radiative Decay (IRD)...'. I think the work is interesting and should be published, but the authors need to cite the existing literature, put their own study into the appropriate context, and they cannot make the current claim of novelty. I therefore don't think that Nature Communications is the appropriate journal for this work.

Version 1:

Reviewer comments:

Reviewer #1

(Remarks to the Author)

The authors have provided, in their detailed response, sufficient evidence that the observed process has general applicability to solvated cationic and anionic species, and they have partially improved the manuscript accordingly. I consider the article potentially publishable in Nat. Comm., but the presentation and discussion require further revision.

I would appreciate if some of the discussion concerning the general applicability of the findings were included in the main text, rather than being confined to the response to reviewers. For example, it should be clarified under what conditions the features of the central ion and the solvation shell can be considered sufficiently isolated, and it should be emphasized that the method applies not only to cationic but also to anionic species.

Regarding the novelty of the work, which reviewer 4 has questioned, I would frame it as follows. There is no doubt that such a process exists, as it is consistent with quantum mechanics and has been observed in MOn species and in similar radiative and non-radiative probe studies. The novelty of the present work lies in demonstrating that the process is detectable in relatively weakly bound systems even at synchrotron facilities, and that the resulting signal contains sufficient structure to potentially yield more detailed insight. Otherwise, stating that the intensity of one-electron transitions is closely related to the overlap of the involved orbitals is merely a restatement of the locality of the dipole operator, which is a trivial observation. Moreover, the authors stress that this is the first such observation for a liquid system. I see no compelling reason why MnO₄²⁻ or CrO₄²⁻ ions in solution should behave fundamentally differently than in solids, as studied decades ago. Therefore, the claimed novelty should focus on solvation-shell-specific aspects rather than the aggregation state.

Another point concerns the repeated distinction between intra- and intermolecular transitions. In this context, that distinction is artificial, since there is no well-defined threshold for interaction strength or bond distance that separates the two regimes. As such, the novelty claims must be carefully reformulated; my suggestion above may serve as a starting point.

In the discussion of orbital composition and charge distribution, and throughout the article, the authors use language that suggests theoretical confirmation of the mechanism. Given that there is no plausible alternative explanation, this is not a confirmation but rather an interpretation. In my view, the explanation is rather straightforward, and I question the need for such detailed discussion in the main text. While this is a matter of taste, the "confirmation" narrative should be avoided.

The conclusion statement: "The IRD spectra are shown to reflect fundamental properties of the solvation shell, e.g., its radius, composition including possible ion pairing, electronic structure, and orientational disorder." should be removed. These aspects are not addressed in the main text, and the supporting information only hints at, without providing conclusive analysis, that such properties might be inferable in future studies. Moreover, other well-established techniques can probe these properties more directly. As I noted in my initial review, merely observing a spectral shift does not equate to gaining

insight. The energy shift is an integral quantity that depends on many variables, and drawing specific conclusions would require extensive correlation studies supported by well-resolved spectral features.

In some places, the structure of sentences and paragraphs does not follow logical flow, implying unwarranted connections between unrelated ideas. For example:

Line 68: "Electronic structure is responsible for the chemical interaction" is too colloquial for scientific writing. Moreover, the second part of the sentence does not logically follow from the first.

Lines 95–100: The authors state that the solvation shell of Na^+ and Mg^{2+} consists of six water molecules, followed by an illogical implication about the energetic position of radiative decay features relative to Kalpha.

Additionally, the final two paragraphs on page 3 should be swapped: first introduce the general scope of the research, then discuss the details. As currently written, the logical flow of that section is unclear.

I also do not share the authors' enthusiasm for repeatedly describing the probe as "from within." What would the alternative be, probing "from outside"? Would that yield fundamentally different information? What these types of processes probe is the local electronic structure, again due to the locality of the transition operator.

It should be explicitly stated that the "HF calculations" are in fact DeltaSCF HF calculations in the main text.

Reviewer #2

(Remarks to the Author)

I appreciate the author's thoughtful response to the original review. Unfortunately, the changes that have been made to the manuscript, especially regarding clarifying the computational details, are still insufficient to merit publication in the manuscript's current form. More specific comments can be found below:

1. Fig. 4 has been visually improved. However, the author's conclusion that IRD is a powerful technique to examine the electronic structure of the surrounding solvent still seems dubious. For Na, it is difficult to visually tell a huge difference between the bulk liquid and the experimental IRD spectrum. Furthermore, the vertical lines from reference 14 and theoretical calculations in this manuscript, do not seem to show good agreement with the observed experimental spectrum. For Mg, there does seem to be a bigger difference between the bulk liquid and IRD spectrum, but once again there seems to be poor agreement with the theoretical results making interpretation difficult.

2. The author's make a seemingly strange or incorrect comment regarding the timescale of the IRD process in response to my original comments as well as in response to a comment from reviewer 1. Specifically, they state that the IRD process probes the core-hole lifetime (on the order of a few fs), but that the IRD process occurs on a timescale of IRD is on the order of ps. These statements seem contradictory. The timescale of the core-hole lifetime should be given by the timescale for the processes that lead to core-hole decay. Therefore, if the core-hole lifetime is on the order of fs, then the core-hole should be decaying via a process that occurs on the fs timescale (such as Auger-Meitner decay or some other more complicated process such as ICD or ETMD). If IRD occurs on the ps timescale, then the core-hole would need to exist on the timescale of ps, which is not the case. Therefore, unfortunately, this reviewer still seems somewhat skeptical about how IRD is physically feasible in this system based on these comments.

3. The only comments to the types of calculations in the analysis and discussion are still only with respect to the Hartree-Fock (HF) approximation. However, the calculation details are still with respect to RAS calculations. Therefore, the same confusion from the original form of the manuscript remains; there is a complete discrepancy between the calculation details and the results presented in the manuscript. What is needed is very concrete statements, such as the results in Figure 1 are obtained from explicitly this type of calculation etc. etc.

4. Though the authors comment that a change has been made, the phrase radiative decay rates is still used in the computational details section.

Reviewer #4

(Remarks to the Author)

The authors have addressed most of the comments, but I would like to see two important changes to the modified manuscript. If the authors make these changes, I support publication.

1) The paragraph "Radiative transitions involving a core hole on one atom and a valence orbital mainly located on another atom....probing chemically and biologically relevant systems in an aqueous environment." needs to be moved to the introduction, as the distinction is important background information for the reader and needs to be addressed early on.

2) Some additional clarification/correction in this paragraph is needed. It is not true that the inter atomic transitions previously observed only occur in covalently bonded systems, they also occur for ionic bonding, such as in MnO for example. The authors should also add the word 'inter atomic' to the description to clarify the distinction between intra molecular (but inter atomic) and inter molecular. This is important as regular XES is always intra molecular (and generally intra atomic).

After moving the text to the introduction, the paragraph could be modified to something like this (plus consistency of reference numbers and changes to the flow of the narrative):

“Radiative transitions involving a core hole on one atom and a valence orbital mainly located on a neighboring atom have been well known for a long time. In transition metal-oxygen ligand systems, such as MnO_4^- , CrO_4^{2-} and VO_4^{3-} , inter-atomic radiative decay has been observed from orbitals primarily located on the ligand oxygen atoms into core-holes centered on the metal ion, see, e.g., [18–23]. Compared to the ions studied here, which form much weaker bonds with the surrounding water, the systems studied in Refs. [18–23] are held together more strongly either by covalent bonds in molecules, or ionic/covalent bonds in the crystalline systems. These inter-atomic transitions observed for the strongly bonded molecular systems are still within the same molecule and can therefore be characterized as intra-molecular. IRD, on the other hand, is inter-molecular, opening up the possibility of probing chemically and biologically relevant systems in an aqueous environment.”

Version 2:

Reviewer comments:

Reviewer #1

(Remarks to the Author)

The authors have adequately addressed my major comments, and I consider the article suitable for publication in Nature Communications.

Reviewer #2

(Remarks to the Author)

The authors have made a substantial effort to update the manuscript. Given the discussion and the changes made I can support publication following the final minor revisions:

1. As far as I can tell, the main text does not contain any results using the RAS calculations. Consequently, the computational details associated with the RAS calculations should be moved to the corresponding section of the SI that uses RAS. Instead, the computational details section in the main text should focus on the data presented in the main text to not confuse the reader.
2. The details with respect to the calculation of the XES spectra in terms of the HF approximation should be more clearly emphasized and made into its own paragraph. These calculations are the main ones utilized in the main text and therefore need to be the most obvious and clearly stated.
3. On lines 312 and 319, the authors should specify which sections of the SI they're referring to.

Reviewer #1 (Remarks to the Author):

The article by Söderström et al. reports the first observation of intermolecular radiative decay in aqueous Na⁺ and Mg²⁺ ions, where a 1s core hole on the metal ions is refilled by an electron from nearby water molecules in the first solvation shell. The authors provide a theoretical analysis of this mechanism and claim that this newly identified effect offers valuable insights into the electronic structure of solvated molecules. While I acknowledge the pioneering nature of this observation and the initial analysis, I have concerns regarding the method's transferability to other systems, the depth of the analysis, and the article's structure and style. The discussion in the main text is overly simplistic, with key details relegated to the supplement, making the main text uninformative when read on its own. Combined with certain technical issues, I find the arguments insufficiently convincing to justify publication in Nature Communications at least in its current form. My detailed comments follow below.

We are delighted to see that the reviewer “acknowledge the pioneering nature of this observation and the initial analysis” and we are very happy to see the overall positive view on our research. There are a few key points raised, and we would like to take the opportunity to discuss these points below. We do hope that with the added information available to the reviewer here, in combination with the changes made to the manuscript, will fully convince him/her as to the general applicability of the IRD method.

The authors present the observed effect as broadly applicable. Indeed, it appears better suited than interatomic Coulombic decay (ICD) for probing the electronic structure of solution bulk. However, its general applicability is questionable. For the phenomenon to occur, the occupied frontier orbitals of the solvent must lie significantly higher in energy than the metal's occupied 2p orbitals to isolate non-local signatures from local metal decay. This naturally occurs in the studied systems due to the highly stable Ne-like configuration of the metal ions. One might expect the effect to be relevant in cases of high ionicity and positive charge, but is it transferable to other systems? In more covalently bound environments, these orbitals approach near-

degeneracy, preventing clear signal isolation. For instance, in transition metal ions probed at the L-edge, the local d orbitals are nearly degenerate with the solvent orbitals, making separation difficult. Additionally, in anionic systems, the signal could appear below the local decay. Does this limit the method's applicability? Can it effectively probe solvation in neutral or merely polar (non-ionic) solutes? These questions require clarification, and the claims should be adjusted accordingly. As it stands, the method does not unequivocally "offer a chemically selective way to probe the electronic structure," contrary to what is stated in the article (see my next comment).

The reviewer questions the general applicability of IRD and we will here discuss this. As IRD is a new tool, its limitations are clearly not well investigated, but we believe that IRD is a general phenomenon in weakly bonded systems, such as liquids. We address the aspects mentioned by the reviewer in more detail below.

Transition metals and anions: The reviewer asks specifically about transition metals; here we are happy to say that we have, after the submission of this manuscript, studied this where we see a clear IRD signal from Cu^{2+} in water. We have also studied anions (F^-) where we observe a clear IRD feature below, and even partially overlapping, the local decay. These preliminary observations confirm that IRD is also present under such conditions, though detailed analysis is ongoing and beyond the scope of the current work. A preliminary version of these spectra are shown below, the analysis is still ongoing and especially for the transition metal seems to be far from trivial. These followup studies will yield results beyond the scope of this paper where we showcase the IRD mechanism. Once the analysis is complete, they will be published separately, further showing the applicability of IRD.

General and energy separation: We would like to thank the reviewer for a very thoughtful comment about the general applicability of the method. We do agree with the reviewer that IRD "appears better suited than interatomic Coulombic decay (ICD) for probing the electronic structure of solution bulk". IRD (as all methods) clearly have limitations, and we will discuss them here, showing that the concerns of the reviewer are unfounded. The main limitation of IRD is that it requires a difference in the energies of the HOMO of the solvent and solute. If these overlap too much, disentangling the two signals will be challenging, but can be done with the help of theory. One such example is shown in the IRD spectrum from F^- below where one of the IRD peaks is partially overlapping the emission from F^- . Thus, the energetic separation does not need to be large, although large separations clearly aid in the identification. This

experimental evidence shows that IRD is a powerful spectroscopic tool even in difficult cases such as an anionic species in aqueous solution.

Regarding the new IRD spectra included in this reply: In short the Cu^{2+} spectrum shows the elastic scattering at 0 eV energy loss, dd-excitations at 0.2 — 2 eV energy loss and IRD features involving water at 2 — 12 eV energy loss. The latter energy coincides with what is typically labelled the charge-transfer band in cuprates.

The emission spectrum from F^- shows the $2p \rightarrow 1s$ emission at ~677 eV and a low energy feature. Overlaid with the experimental spectrum is both the water valence UPS spectrum, as well as preliminary DFT calculations, these are both shifted to align with the peak at ~671.5 eV.

Covalently bonded systems: We would like to note that IRD by definition cannot be observed in covalently bonded systems. In these systems the decay is always *intramolecular* and never *intermolecular*. As pointed out by reviewer #4 (see below) *intermolecular* transitions, even when the partial density of states is significantly located on one of the ligands, have previously been observed. In the case of IRD all transitions have to be between two separate systems. We do acknowledge that the border between separate systems and molecular complexes is not trivial to define, but it is clear that there is a difference between strongly bonded molecular systems, such as those exemplified by reviewer #4, and ions dissolved in water – as discussed in this manuscript. For a discussion about covalently bonded solutes see below.

Neutrals and polar solutes: As for probing neutrals and polar solutes (e.g. alcohols and amines) we can only speculate at present; however, we have a long-term goal to investigate this as well. As long as the overlap between valence orbitals of the solvent and solute is non-zero IRD will occur, the question is if we can detect it. We have preliminary measurements from the ammonium ion in water where IRD is observed – however for molecular systems the spectra become more complex and theoretical input clearly aids the interpretation. Such an in-depth study is clearly beyond this paper describing the initial finding of IRD.

To summarize: By complementary measurements of transition metals, anions and also molecular ions we have shown that IRD is not limited to neon-like ions but appears to be a general phenomenon. We have outlined the limitations we are aware of today and we hope that this is a sufficient answer to the questions raised. We do acknowledge that there are still remaining questions to be answered, however we believe these are beyond the scope of this paper describing the initial finding.

Changes: We have not made any changes to our manuscript based on this question. However, we hope that our answers satisfy the reviewer.

Figure 1: (Left) A zoom of emission from F^- in water (red). The zoom is to highlight the IRD features. The IRD features can be fitted using three gaussian components, even without the input of theory. The black curve shows the liquid water valence band as in the submitted paper, and the blue curve shows preliminary theoretical calculations. All curves are aligned to the low-energy peak in both energy and intensity. See text for further details. (Right) The energy loss spectrum of Cu^{2+} in water excited at the L_3 edge, showing both elastic scattering, dd-excitations as well as IRD, see text for details. The inset shows a zoom highlighting the IRD features. Here it is worth noting that the IRD features cannot be well described using three gaussian components but needs at least 4. We are in the process of evaluating this new data.

The authors claim that the method is well suited for probing electronic structure. However, they experience difficulties in drawing clear conclusions due to overlapping spectral bands and in explaining energy shifts and intensity variations. If such complications arise in these relatively simple systems, how can the method be reliably applied to more complex cases? A rigorous analysis, such as for an octahedral solvation shell, would involve distinguishing different groups of water orbitals. Based on symmetry considerations, the $2a_1$, $1b_1$, $3a_1$, and $1b_2$ orbitals should form distinct bands, split into singly, doubly, and triply degenerate groups according to octahedral symmetry representations. Hydrogen-bonding lone pairs ($3a_1$ and $1b_1$) would likely show the most significant energy splitting and potential overlap. The conclusions on electronic structure should also address nuclear dynamics-related factors, such as the statistical distribution of first-shell water molecules and contributions from contact pairs. While theoretical studies on these aspects are extensive, the main text merely mentions them, relegating details entirely to the supplement. If such

analysis proves difficult even for these simple systems, the claim of providing “an insider’s view of the solvation shell” should be significantly tempered. Moreover, as the study examines electronically relaxed species post-ionization, it remains unclear how this relates to the original solvated state.

Again we would like to thank the reviewer for an insightful comment. In order to answer this we would like to break this down into several parts.

- a) Drawing clear conclusions: Here we do not agree with the reviewer, these energy shifts and relative intensities are discussed in the sections “The IRD mechanism” and “IRD and the electronic structure” of the main manuscript, and further elaborated on in the SI. As shown elsewhere in this reply we have recently observed IRD for significantly more complex systems, showing that our general framework is more generally applicable.

We would also like to point out that the measurements serve as validation tests for any type of calculations that are carried out - thus good agreement enhances the value of the informational content of the calculations for prediction of, e.g., geometrical and electronic structure, which then can serve also as basis for interpretation and structure - property relations, and vice versa - poor agreement reduces such value.

- b) Symmetry considerations: The reviewer is clearly correct in this, albeit the exact interpretation is somewhat more complex. In Fig. 2 of the main paper several peaks are predicted by the theoretical model, and most are observed in experiment. In a first approximation these can be associated with the $2a_1$, $1b_1$, $3a_1$, and $1b_2$ orbitals. In our experiment we focused on the three outer valence orbitals that are close in energy, however the theoretical model clearly shows the existence of a band associated with $2a_1$, due to time-constraints this was never attempted to observe experimentally.

As a starting point, we have considered octahedral symmetry for which such relatively simple orbital assignments can be done. We have used this idealized structure to clarify the mechanisms underlying IRD. In the SI we have discussed and shown how the various orbital contributions make up the IRD spectrum in an idealized geometry. This shows how the outermost orbitals of liquid water further splits in energy, exactly as the reviewer mentions. However, the first solvation shell is highly dynamic in nature, not exhibiting one single structure. We discuss the dependence on distance, orientation, and composition in the main manuscript, and further elaborate on this in the SI. We do hope that this sufficiently answers the questions raised by the reviewer. In short, we fully agree with the reviewer and we also believe that we have shown this in our

work. However, as this evidently was not clear we have revised our theoretical presentation, which is also in line with other suggestions.

- c) Contact ion pairs: We do agree with the reviewer that this is indeed an interesting subject. For the investigated cases, ion-pairing is predicted by MD to be very weak, see e.g. Ref. 14 in the manuscript, in agreement with our non-observation of IRD signals associated to ion pairs. We have modelled the IRD signal from contact ion pairs, however this study focuses on the existence of IRD as such. We do agree with the reviewer that this must be explored, and we have applied for experimental time to do so, however this is clearly outside the scope of this paper describing the first observation of IRD.
- d) Time-scales: The reviewer is indeed correct in pointing out that IRD involves final-states with electronically relaxed species. The intermediate state can be both ionized as well as excited, but the point remains valid.

We are not sure that we understand the reviewer's concern. We agree that electronically relaxed species post-ionization are important for an understanding of the spectroscopic results. Like in all spectroscopies, IRD probes differences between states. From these differences, information on e.g. the electronic structure of the ground state is obtained.

We like to point out here that the radiative core-hole decay follows the final state rule (as is both rigorously derived and tested many times earlier). That means that if one set of orthogonal orbitals is to be chosen to interpret an X-ray emission spectrum - it is the ground state orbitals (as we also use in the MD).

The IRD process occurs on the core-hole lifetime scale, i.e. a few femtoseconds. On this short timescale nuclear motion is negligible.

- e) Main text vs. SI: As mentioned above we made the active choice to have the main text on a level for a general audience, with more details presented in the SI. However, we do see that several reviewers pointed out that the theoretical part was hard to follow - we have thus made overall changes to the manuscript where we try to improve this.

Changes to the manuscript: We have moved part of the theoretical results from the SI into the "The IRD mechanism" section of the main manuscript, in order to enrich the discussion there.

Regarding the computational setup, the discussion of Hartree-Fock (HF), RASSCF, and DFT calculations is somewhat disorganized, making it unclear which method was used for which aspect of the study. From the explanations, it appears that what the authors refer to as "HF" is actually a configuration interaction singles (CIS) method with orbital optimization enabled via RAS subspaces. Otherwise, how could excited single-hole states be computed with HF? Additionally, the rationale for using DFT to compute intensities is unclear when OpenMolcas could handle them directly, even incorporating perturbation theory to improve correlation effects. Concerning correlation, the active space could be expanded to include unoccupied orbitals, such as the 3s of Na or Mg and two antibonding water orbitals, to capture additional correlation effects. Even more crucially, including the metal 3p orbitals would provide a stronger theoretical foundation for mechanism 2 on page 6 of the supplement. While correlation (both RASSCF and RASPT2) may only subtly alter intensities by mixing configurations with electrons in the metal's unoccupied 3p orbitals, such adjustments could be decisive given the already weak transition intensities. The authors also briefly mention vibronic effects, but these should significantly impact both intensities and energy shifts due to substantial structural differences between $M^{(q+1)}-H_2O$ and $M^q-H_2O^+$ species.

Regarding HF (but the same goes for DFT) we recall that it is possible to optimize states with a hole (either in the core or in the valence shell) by imposing that an orbital always maintains a single electronic occupation during the self-consistent optimization procedure. This is the ground for the Delta-SCF technique which has been commonly applied for decades. There is no need to invoke any configuration interaction because each hole state is, in the present case, a spin doublet which simply can be described by a single determinant, i.e. as in the approximation of independent particles. A comment about this has been inserted in the SI.

The use of HF and DFT is motivated by the computational simplicity and therefore by the possibility to easily perform calculations on large systems and for different geometries. This has been functional in investigating the dependence of the IRD bands on water molecules in the second solvation shell or for different structures of the clusters deriving from MD or for idealized geometries such as those with "tilted" water molecules. Calculations with RASSCF+RASPT2 have also been done, even if they are computationally more complex, for the specific idealized model to compare

with the HF and DFT calculations. Our conclusion was that the independent particle approximation was more than sufficient to interpret and predict the IRD process. Thus tilted calculations with RASSCF / RASPT2, that consume huge computer time, were not considered motivated by limited additional information that they possibly would provide.

Concerning role of vibronic effects we remind that according to the Born-Oppenheimer and Franck-Condon principles, vibronic interaction will not change the band intensities, nor shift the energy of the electronic bands, meaning that the bari-center of the electronic bands will remain the same, while there might be a very tiny, and for all practical cases negligible, shift in the band maximum position. Thus, the effect of vibrations will only be to broaden the bands and not to shift energy nor change the total intensity, then not affecting the overall picture of the IRD process that we are proposing.

We agree that inclusion of M 3p, 4p, and water virtual orbitals in the active space is an issue. However, based on our main goal to have a model for the IRD/ K_α relations, and also the fact that good comparison with the experiment is obtained a posteriori, we believe that our active space is a good choice compromising accuracy versus efficiency of calculations.

The analysis of Mulliken charges as an attempt to explain intensity borrowing is misleading. Transition intensities depend on dipole matrix elements, not charge distributions. The correct explanation appears only in the supplement, though it focuses on orbitals rather than many-electron states. Additionally, the mention of the one-center approximation in the context of a non-local effect is confusing. While the supplement clarifies the authors' intended meaning, the main text's brief reference is insufficient and could be misinterpreted. Furthermore, the statement that "the intensity and the electron come from different sources" is poorly worded and misleading—intensity and electron transfer are distinct concepts that should not be conflated.

Overall, the main text contains redundant and overly simplified explanations (e.g., the section "The IRD mechanism," particularly the paragraph following line 204), making the intended audience unclear. Meanwhile, most of the substantive discussion is confined to the supplement. Ideally, this should be reversed: the

main text should provide the core scientific discussion, with supplementary material used for additional details. As it stands, the main text simply lists what was done, whereas the supplement contains the actual scientific discourse.

We want to remind the reviewer that the local charges (Mulliken or not) have turned out to be excellent tools to interpret various types of X-ray spectra - XES, XAS, Auger, valence XPS, as they form the basis of so-called one-center approximations. These approximations are rigorously derived from the total transition moment in the form of Fermi's golden rule.

The one-center approximation in the context of a non-local effect is hopefully not confusing any longer, we only have a different type of application than the usual, here the intensities are guided by a relatively small contribution from the metal local, one-center, part of the wave function (or orbital), while the main part of the transiting charge originates from the non-local waters. Adopting quantum mechanics this is not strange, only unusual, something that actually makes our discovery of IRD even more interesting.

Changes to the manuscript: As suggested at several points we have re-structured and somewhat re-written our paper to make the theoretical framework clearer for any audience. We have also clarified the sentence "the intensity and the electron come from different sources" which now should be clearer.

- a) Mulliken charges: The statement "*The analysis of Mulliken charges as an attempt to explain intensity borrowing is misleading*" is probably due to a misunderstanding. The Mulliken charge analysis was used to illustrate the charge redistribution during IRD, not to infer transition intensities. As clarified in the revised manuscript, intensity arises from dipole matrix elements, and the charge analysis provides complementary insight into the non-local character of the process. The main point with Fig. 3 is to describe how the charge is localized/delocalized in various steps of the IRD process. We agree that "*Transition intensities depend on dipole matrix elements*", but also like to point out that charges have frequently been used in the family of X-ray spectroscopies to approximate and analyse the full dipole transition matrix elements. To clarify this further, we have moved the theoretical discussion describing this from the SI into the "The IRD mechanism" section of the main manuscript. We believe this has enriched the discussion there, which will be helpful for the readers.
- b) One-center approximation: We do see how this can confuse a reader and have changed our manuscript to ensure that this is clearly understood. We have

moved part of the theoretical results from the SI into the “The IRD mechanism” section of the main manuscript, in order to clarify the role of the one-center approximation in the context of a non-local effect.

- c) “the intensity and the electron come from different sources”: We do see what the reviewer means with this and have added the following sentence to our manuscript to clarify this statement. “While the electron originates from the water molecules in the first solvation shell, the IRD signal strength (the intensity) originates from the hybridization with the metal valence-orbitals.”
- d) Main text vs. SI: See reply above, in short this is our attempt at making the main text appealing to a broader audience and to refer the experts to the discussion in the SI.

Minor:

- In the introduction, the paragraph beginning “The electronic configuration of the free metal...” is too abrupt. Why were these two chosen? Why is the charge $q+1$? Basic explanations from elsewhere in the article could be relocated here.

In the ground state, the metal ion M has charge q ($+1$ for Na and $+2$ for Mg). Core-ionization removes one electron, resulting in charge $q+1$. The radiative decay does not change the total charge, so also the final state has total charge $q+1$.

Changes: We have added a sentence and re-written parts of the text to clarify what this means.

-
- Lines 118–119 discuss the 2p orbitals of free ions, yet the conclusion pertains to solvated ions.

The metal 2p orbitals are typically treated as core orbitals that are little affected by the surrounding. In the analysis we explicitly discuss the influence of solvation, in terms of small shifts and hybridization. The effects are so small that it is appropriate to still refer to them as metal 2p orbitals.

Changes: We have not changed the text since we do believe that our presentation is correct.

-
- The caption of Figure 2 mentions a shift in computed spectra but does not specify the value.

As the calculations often provide a reasonable accuracy for the relative energy scale, the absolute scale is often not correct. This shift is only to account for this effect as we do not consider that it has any physical meaning, thus we have chosen to omit it.

Changes: None as discussed above.

- In line 217, orbitals are referred to as “metal-centered” despite being described as having “small metal character.”

Changes: We have changed the text describing this and we now hope that this is clear.

- In line 242, emission energies are linked to electron binding energies; however, ionizing and non-ionizing transitions are not equivalent processes.

This is absolutely correct. In the presented data the first step is an ionization of the metal ion. We here point out that the energy of an X-ray emission transition is exactly the same as the difference between a core electron binding energy and a valence electron binding energy, which has been utilized frequently in the past.

Changes: We have not changed the text as we believe what is written is correct.

Reviewer #2 (Remarks to the Author):

This article involves a joint experimental and theoretical investigation of a new process known as intermolecular radiative decay (IRD) occurring for metal ($\text{Na}^+/\text{Mg}^{2+}$) ions in water. The process involves initial ionization of the 1s core-hole state of the metal ion, which induces relaxation of a valence electron residing in the solvation shell and emission of a photon. The hypothesis is that this process can be used as a useful probe of the local solvation shell.

Unfortunately, there are some key limitations both in the quality of the data and in the structure of the paper, which makes it difficult to recommend the paper for publication in Nature Communications as outlined below:

- 1. The key experimental findings of the manuscript are associated with the results presented in figure 2 and 4 associated with the spectra assigned to IRD. The author's claim that these spectra can be used as a useful tool to examine the electronic structure of the solvation shell local to the metal ions in comparison to bulk water. However, the noise observed in the spectrum, especially for Na^+ spectrum (associated with the oscillations) seemingly makes it impossible to distinguish if any features change from the bulk structure. Even the Mg spectrum appears to have very sharp features, for example the very pointed peak for the 3a1 part of the spectrum, which would seem to indicate that only a few points are used to plot the spectrum. To draw any conclusions, it would be necessary to plot some form of error bars on the spectrum. It would also be beneficial to plot a difference between the bulk and the IRD spectrum as well.*

We do agree that Figs. 2 and 4 show the experimental result, however we do not share the reviewer's view on the level of noise. We have re-made Fig. 4 in order to show the data-points and overlap this with a fit. This new figure should show that any sharp features are only associated with single points and should thus be seen as noise. Hopefully this makes the data-quality clearer, this also shows the number of datapoints used to plot the spectra. The difference in datapoints is due to different experimental conditions for the two samples. This new figure hopefully shows that what the reviewer calls oscillations in the Na^+ spectrum is nothing but noise and cannot be interpreted in any way. This statistical noise is clearly visible in the parts of the IRD spectrum

lacking signal, from these parts of the spectra (especially for Na-case) it is clear that the observed “oscillations” is only statistical noise. In the new figure it is evident that the pointed peak for the 3a1 part of the spectrum in the case of Mg is only one point and should not be interpreted in any way except statistical noise.

The creation of a O 1s core hole in liquid water induces intramolecular dynamics that heavily affects the spectrum, and it is considered to be the principal reason for the apparent split of the sharpest peak in the spectrum [see e.g. <https://doi.org/10.1073/pnas.1815701116>]. In IRD, no such dynamical effects due to the O 1s core hole are present. The dynamics associated with creation of a core hole on the neighbouring ion is entirely different, and much less intramolecular vibrational excitations are to be expected. Therefore, the comparison to the photoemission spectrum of water is more appropriate in our case, as this probes the single-hole final states without any core-hole-induced effects. To avoid a lengthy discussion of the dynamics in connection with the XES spectrum of bulk water, we have chosen to omit this comparison.

Related to error bars we do not believe that these are relevant in any of the figures of the paper. Any reader will easily be able to judge the quality of the fit by eye. As the fit is only used to extract a total area of the IRD feature we are of the opinion that these error bars would not help with the interpretation.

Changes: We have re-made Fig. 4 to clearly show what part of the signal is statistical noise.

2. The authors do not hypothesize nor discuss what other processes may be contributing to the observed signal. For example, is it possible to have resonant hole transfer to the solvation shell to generate an excited ionized state of the neighboring solvation shell, which can then subsequently decay?

The reviewer is pointing out an interesting mechanism with the idea that the core-hole would transfer from the metal-ion to the water. We did indeed not consider this pathway as this cross-section is negligible. However, even if this process would occur frequently the emitted X-rays from ionized water would be detected in an energy-window around 520 – 530 eV. This would not contribute to our IRD signal which is detected at significantly higher photon energies. This process would also be indistinguishable from direct ionization of the water molecules, but interestingly it would emphasize the first solvation shell. We have made calculations of such “core-hole jump processes” in the case of the isolated CH₃Cl molecule (between C1s and Cl1s) and the calculated oscillator strengths are extremely small. To the best of our

knowledge such “core-hole jumps” have never been observed before and would likely require experiments at XFELs in order to be observed.

Changes: None

3. Given that it is known, and the authors even state, that ultrafast non-radiative processes such as Auger, ICD, and ETMD occur in these systems, it is surprising that any radiative process is observed. Are the authors able to make any comment to the relative timescale of the IRD process in comparison to the other known competing processes? This would help solidify why IRD is even feasible in these systems.

We absolutely agree with the reviewer, it is surprising that this radiative process is observed. This is one of the reasons we believe that this discovery is of interest to a broader audience, and why we submit this finding to Nature Communications. Given the present experimental data we can estimate the IRD time-scales from the observed IRD/Kalfa intensity ratios relative to the lifetime of the Na and Mg 1s core-hole lifetimes, ≈ 2.3 fs and 2.0 fs, respectively (see e.g. J.L. Campbell and T. Papp, Atomic Data and Nuclear Data Tables 77,1 (2001)). This lifetime is mainly determined by the decay rate of the dominant KLL Auger-Meitner decay. The radiative fraction of the $1s^{-1}$ core hole decays, mainly the Kalfa line, is $\approx 2.1\%$ for Na and $\approx 2.8\%$ (see e.g. Krause, M. O, J. Phys. Chem. Ref. Data 8, 307 (1979)), giving the time scale of Kalfa radiative decay as ≈ 109 fs for Na and ≈ 71 fs for Mg . The intensity of the IRD line is $\approx 1\%$ of the Kalfa line for Na, and $\approx 1.5\%$ for Mg, which gives an IRD timescale of ≈ 11 ps for Na and ≈ 5 ps for Mg. The time scales of the core hole decay processes thus range from the low femtosecond timescale for KLL Auger-Meitner to a few ps for IRD, spanning almost three orders of magnitude. However, we would like to stress that this does not mean that IRD probes a longer time scale than Auger-Meitner. Instead, all the core-hole decay processes probe the same time scale, set by the 1s core-hole lifetimes. We have therefore chosen to not include this discussion in the manuscript.

Changes: None

4. Figure 3 presents the Mulliken charges for the various states for Kalfa decay vs IRD. The authors make conclusions about the nature of charge-transfer for IRD based on these results. However, it seems like that this is the only possible conclusion

that could have been drawn given that the authors are specifically choosing what the initial and final states are along the pathway. i.e. they are choosing that the IRD final state specifically corresponds to one involving the electron density on water. Now maybe this isn't the case, but the calculation details are so limited in scope (see point 5) that it's not possible to tell if this is a trivial result.

In Fig. 3 we focus our attention only on the radiative decay channels in order to contrast the local (K_{α}) to the non-local (IRD) decay. Here we ignore all other decay channels such as Auger-Meitner decay, ICD etc. The main idea with including this figure is to show how the charge density is (de)localized in various steps of the IRD process. Fig. 3 does not attempt to describe all possible decay mechanisms, as such a figure would be much more complicated. As the experiment only shows the IRD signal, the electron delocalization process during IRD is what we want to show in this figure.

Changes: We have added the words “for the radiative decay of the core hole” in the figure caption to clearly indicate this.

5. The presentation of the calculation details is both disorganized, appearing separately in the methods section as well as various sections of the SI, and without sufficient detail to understand how the calculations are being performed. The paper needs to be reorganized such that its clear what calculations are being exactly performed and what is the connection between all the seemingly disparate calculations. For example:

a. The only comments to the type of calculations used in the analysis in the results and discussion section seems to be the phrase “X-ray emission spectra for Na⁺ and Mg²⁺ ions within a cluster of six water molecules have been calculated in the Hartree-Fock (HF) approximation” on page 4 and “Figure 4 also shows calculated IRD spectra, obtained by HF calculations for ...” on page 8. However, the methods section provides details on RASSCF and DFT calculations, which does not match the comments about HF in the results and discussion section.

Regarding HF (but the same goes for DFT) we recall that it is possible to optimize states with a hole (either in the core or in the valence shell) by imposing that an orbital always maintains a single electronic occupation during the self-consistent optimization

procedure. This is the ground for the Delta-SCF technique which has been commonly applied for decades. There is no need to invoke any configuration interaction because each hole state is, in the present case, a spin doublet which simply can be described by a single determinant, i.e. as in the approximation of independent particles. A comment about this has been inserted in the SI.

The use of HF and DFT is motivated by the computational simplicity and therefore by the possibility to easily perform calculations on large systems and for different geometries. This has been functional in investigating the dependence of the IRD bands on water molecules in the second solvation shell or for different structures of the clusters deriving from MD or for idealized geometries such as those with "tilted" water molecules. Calculations with RASSCF+RASPT2 have also been done, even if they are computationally more complex, for the specific idealized model to compare with the HF and DFT calculations. Our conclusion was that the independent particle approximation was more than sufficient to interpret and predict the IRD process. Thus, tilted calculations with RASSCF / RASPT2, that consume huge computer time, were not considered motivated by limited additional information that they possibly would provide.

b. The authors make a statement that RASSCF was used to calculate radiative decay rates. However, there was no discussion or comment made in the main text about the decay rates.

“Radiative decay rates“ have now been changed to “radiative transition intensities”, but we note that the two concepts are related.

We recall that we use RASSCF/RASPT2 calculations for all spectra calculations for $n=1,4,6$.

i) We performed 'RAS' calculations with only one single determinant for the (closed-shell) initial ground state, which is exactly the same as a HF.

ii) We did 'RAS' calculations for the M1s core-hole state, which was also a single determinant wave function (equivalent of a ROSHF - Restricted Open Shell HF wave function).

iii) We did a RAS calculations for the final states considering a single hole on all possible valence orbitals. In this case the wave function is not a single determinant and all states were allowed to interact via CI, and the orbitals have been state-averaged in the optimization. So this is not a HF or ROSHF.

Changes: We have made a clarifying remark of this in the text.

c. There is no clear discussion about how the emission energies are obtained. If they're at the RASSCF level, how were the specific ionized states chosen? If they're obtained specifically from a pure HF calculation, is this done by simply looking at the MO energies of a ground-state HF calculation? Significantly more detail is necessary because as of now it would be impossible to reproduce the results.

There are many details in the calculational procedures which are left out, partly because the authors do not think they qualify for a text space, and also because they by now represent standard procedures. Nevertheless, we recall that both in the HF and DFT cases the doublet states (core hole and valence holes) between which the decay occurs are represented in the approximation of a single determinant with occupation 1 in a specific orbital (core or valence). The core hole state has been optimized by fully including the electronic relaxation around the hole. The final valence hole states are also described by a single determinant expanded, however, on the ground state orbitals, i.e. in the frozen approximation, i.e. without including the electronic relaxation around the valence hole. This is generally a better choice for the core valence states because electronic relaxation and electronic correlation (of course not included in the HF approximation) have comparable energy values and of opposite sign and therefore tend to cancel each other. This also follows the so-called final state rule for analysing X-ray emission spectra. The Kohn-Sham orbitals of the DFT calculation also describe the dynamical part of electronic correlation. The decay intensity has been computed in the dipole approximation from the transition moment between determinants, not orbitals.

Changes: An explicit comment about it has been added to the SI.

With the RASSCF method (which describes the different states as combinations of determinants) + RASPT2, the electronic correlation is accounted for in both the core hole and valence hole states. However, the variations in the energies and IRD intensities are rather small and do not change the interpretative framework of the IRD process. The IRD intensity has been calculated, in all the approximations, by the transition dipole moment between core hole state and valence hole states projected, therefore, on different and non-orthogonal sets of orbitals.

d. The analysis performed in section 2.4.1 seems to involve calculating matrix elements between single molecular orbitals.

However, like point c, no explicit comment has been made as to where the MOs are obtained. Are these MOs obtained from a ground-state HF calculation?

The origin of the MOs (HF or Kohn-Sham) was described in the previous comment. Regarding the calculation of the transition moments, it was done between states (single determinant or combination of determinants) and not between MOs.

e. Section 2.4.2 makes comments to a pure HF calculation, but it is unclear how these calculations relate to any previous electronic structure calculations as they use a different basis set. There is also no detail provided as to the specifics of these calculations. Does this involve a pure ground-state calculation? This doesn't seem correct as the authors comment that they are somehow targeting "the relaxed M1s-1 state" which would not correspond to a ground-state configuration.

As explained above, in HF and DFT calculations the core hole state includes electronic relaxation, while the valence hole state does not (frozen orbital approximation).

Changes: An explicit comment about it has been added to the SI.

f. Section 2.5 describes the use of an additional basis set that once again has no connection to the previous sets of calculations.

All results reported were obtained using the aug-cc-pVTZ basis set. The ANO-RCC-VTZP basis set was used for some preliminary calculations, but unfortunately it was mentioned in the manuscript. We thank the reviewer for spotting this mistake, which we have corrected.

Changes to the manuscript: The ANO-RCC-VTZP basis set is not mentioned, only the aug-cc-pVTZ basis set, which is the one used for the results.

g. Additionally, the authors should include the geometries and, if possible, input files of any electronic structure calculation performed to allow for ease of reproducibility.

The geometry data form a quite huge data set. We have now offered the interested readers access to geometries and input files on request.

A few additional more minor comments:

6. The authors should define what q is on page 3 in the notation M^{q+1} i.e. that q corresponds to the original charge of the cation prior to core ionization.

Changes: This section was pointed out by reviewer #1 as well and this has been changed for clarity.

7. How did the authors choose the representative configuration from the molecular-dynamics trajectory?

Representative clusters were extracted from the MD trajectories with a GROMACS tool called "cluster" which groups the frames according to their mutual similarity.

Changes: This is now described in connection to the discussion about the representative configuration in the section "Model system structures" in the SI.

8. The following phrase "The HF approximation, fully including the electron relaxation around the tinial core hole..." on page 13 is strangely worded. Do the authors mean to say if some % of exact HF exchange is included in the DFT functional? Electron relaxation is inherently included in any SCF calculation whether it's DFT or HF.

"Electron relaxation around the core hole" means what occurs when the starting (sudden) state is the ground state from which an electron is instantaneously removed from a core orbital (frozen orbital approximation) and this state is then variationally optimized so as to describe the process by which the core hole "attracts" the electron density around itself.

Changes: Some rewording along these lines has now been made in the SI.

Reviewer #3 (Remarks to the Author):

This article reports on a highly original observation, which identifies X-ray emissions of core-excited aqueous solutions of Na⁺ and Mg⁺⁺ ions lying about the K α emission as a valence solvent to metal core orbital filling via emission of light. To my knowledge, this represents a new type of observable of the solvation shell, which has so far been lacking (or been indirect) in X-ray studies of solvated species. The arguments put forward by the authors and the theory solidly support their interpretation. It would be now interesting to extend such studies to other solutes and eventually, other solvents too, such as alcohols. I strongly support publication without changes.

We do thank the reviewer for his/her enthusiastic reply and view on our research. While we do agree with the general statement we also see the need for some alterations based on the input from the additional reviewers. We also agree with the suggested research directions proposed where some are presented in this reply (other solutes) and others will be pursued in the years to come (even more solutes and other solvents).

Reviewer #4 (Remarks to the Author):

This is a well-written and carefully conducted study. The authors support their experimental findings with convincing theoretical analysis, and the results are presented very clearly. My main issue with this paper is its claimed novelty as stated by the authors 'Here, we report experimental results on a novel non-local X-ray emission process, Intermolecular Radiative Decay (IRD).'

We are again very happy to see that also reviewer #4 finds our manuscript and study to be "well-written and carefully conducted study. The authors support their experimental findings with convincing theoretical analysis, and the results are presented very clearly.". We also see the point the reviewer is making related to the novelty of these transitions. We have at length elaborated on this below and we have also included some clarifying statements and references in the paper.

The phenomenon of non-local x-ray emission described and named IRD by the authors is not new. Already in 1966, Best (Best, Electronic Structure of the MnO₄⁻, CrO₄²⁻, and VO₄³⁻ ions from the Metal K X-Ray Spectra, Chem. Phys. 44 3248 (1966)) and in 1975 Jones and Urch (Jones and Urch, Metal-ligand bonding in some vanadium compounds: a study based on X-ray emission data, J. Chem. Soc., Dalton Trans. 1885 (1975)) assigned weak x-ray emission lines (~ 50 eV above the K β main line) observed in transition metal oxides as ligand 2s to metal 1s 'interatomic' or 'crossover' transitions. Best starts the abstract of his paper with 'New measurements of the K x-ray emission spectra from the metal atoms in MnO₄⁻, CrO₄²⁻, and VO₄³⁻ ions are presented and discussed in terms of dipole transitions from occupied orbitals to the 1a₁, i.e., the metal 1s vacancy.'

A detailed study of the intensity and spectral position of these interatomic emission lines related to the metal-ligand distance has been published in 1999 for a series of Mn oxides (Bergmann et al, Chemical Dependence of Interatomic X-Ray Transition Energies and Intensities - A Study of Mn K β " and K β _{2,5} Spectra, Chemical Physics Letters 302, 119 (1999)). Since then, these transitions have been applied numerous times to identify and characterize ligand atoms (see, e.g. Pollock & DeBeer, Valence-

to-core X-ray emission spectroscopy: a sensitive probe of the nature of a bound ligand, Journal of the American Chemical Society 133, 5594 (2015)), led to an important breakthrough in the structure of the FeMoCo cluster in nitrogenase by identifying the central carbon atom (Lancaster et al, X-Ray Emission Spectroscopy Evidences a Central Carbon in the Nitrogenase Iron-Molybdenum Cofactor, Science 334, 974-977 (2011)), and were employed to directly measure the oxygen ligands in the Mn cluster of the photosystem II protein in a water solution (Pushkar et al, Direct Detection of Oxygen Ligation to the Mn₄Ca Cluster of Photosystem II by X-ray Emission Spectroscopy, Angew Chem Int Ed. 49, 800-803 (2010)). Several other studies have been reported as these transitions have become a powerful tool in x-ray spectroscopy.

We do thank the reviewer for taking the time to point out some literature that was indeed not cited properly in our manuscript. We have revised the manuscript to cite this literature properly and give our view on why the processes are different, something we like to elaborate on here.

The systems mentioned by reviewer 4 are very different to those studied here. The metal-ligand complexes mentioned by the reviewer involve oxygen ions, not water molecules as in our case. The bonding is also very different, mainly weak ion-dipole interaction in our case, and strong polar covalent bonds for the metal-oxygen ligand complexes. On the macroscopic scale, bonds such as those in the metal-oxygen ligand complexes are found in minerals, whereas the water molecules have a mean residence time in the first solvation shell around a solvated ion such as Na⁺ or Mg²⁺ on the order of microseconds or less, see for example L. Helm and A.E. Merbach, Coordination Chemistry Reviews 187, 151 (1999).

When ions are dissolved in water they are discussed as charged ions with a coordination shell of water molecules (the first solvation shell), which is weakly bonded to the ion. While there are several higher order solvation shells, the first solvation shell shows the largest deviation from bulk water. This solvation shell is dynamic in nature with significant variations on both length between the ion and the water molecules, as well as the geometric structure; as discussed in the SI. As these figures appeared only after the references in the SI we do see that they can easily be missed, and we have now amended this. As the reviewer points out, *intra*-molecular radiative decays from systems bonded by molecular complex bonds or covalent bonds have been observed

for decades, here we show the first instance of *inter*-molecular radiative decay from weakly bonded aqueous systems.

While the IRD transitions effectively are described by the same matrix elements as the *intramolecular* radiative decay in the references provided by the reviewer, we do find that there are key differences between the two processes making IRD significantly different from the *intramolecular* “cross transitions” previously observed.

To conclude, we are convinced that IRD will have significant use in order to probe solvation-shells in bulk water, which is of importance for several fields outside physics, hence we believe this to be interesting for a wider audience, as reached via Nature Communications.

Changes: In the section “IRD: Observation and interpretation”, we have added a discussion of the *intra*-molecular transitions in strongly bonded systems brought up by the reviewer. This addition clarifies how IRD, which involves *inter*-molecular transitions between weakly bonded species, is different:

“Radiative transitions involving a core hole on one atom and a valence orbital mainly located on another atom are not unknown. In transition metal-oxygen ligand systems, such as MnO_4^- , CrO_4^{2-} and VO_4^{4-} , radiative decay has been observed from orbitals primarily located on the ligand oxygen atoms into core-holes centered on the metal ion, see, e.g., [18–23]. The systems studied in Refs. [18–23] are very strongly held together by polar covalent bonds, while the ions studied here form much weaker bonds with the surrounding water. The transitions observed for the strongly bonded systems are *intra*-molecular, whereas IRD is *inter*-molecular, opening up the possibility of probing chemically and biologically relevant systems in an aqueous environment.”

While in these examples transition metal complexes are studied, the underlying process is identical to what the authors call IRD as described in the submitted manuscript, namely a 1s core hole of the element that is probed is filled by an electron from a neighboring atom or molecule (in the submitted case the water around the ion, in previous cases ligand atoms of molecules). From my understanding, the treatment for calculating the transition energies and strengths is also the same as in those previously published studies. I therefore do not agree with the claim by the authors in their conclusions ‘Combining experiments and calculations, we have demonstrated the existence of a novel

core-hole decay process, Intermolecular Radiative Decay (IRD)...'. I think the work is interesting and should be published, but the authors need to cite the existing literature, put their own study into the appropriate context, and they cannot make the current claim of novelty. I therefore don't think that Nature Communications is the appropriate journal for this work.

We agree with the reviewer that there are clear similarities between IRD (*intermolecular* radiative decay) and the *intramolecular* radiative decay discussed by the reviewer. It can thus be discussed whether IRD should be regarded as a new process, or the surprising and unpredicted observation of an established process in a very different type of system. We have consequently modified the title and the discussion to reflect this.

As pointed out by the reviewer the *intramolecular* radiative decay has indeed led to breakthrough discoveries for molecules with ionic covalent bonds, with the discovery of *intermolecular* radiative decay presented in this study we believe that similar advances will be made for chemically and biologically relevant systems in an aqueous environment. Furthermore, we do not consider it a weakness that the general theoretical framework describing radiative transitions is applicable also for *intermolecular* transitions in these weakly bonded systems.

Changes: See comment above.

To summarize we would like to thank all reviewers for their effort in reading our initial manuscript and through this process aiding us with significantly improving the quality of our initial manuscript.

On behalf of the authors

Johan Söderström

To the reviewers,

First, we would like to take the opportunity to thank the reviewers for taking their time for a detailed second review of our updated manuscript.

In their present reply, the three remaining reviewers raise some questions, and below we have addressed them one by one. Again, we feel that these changes indeed have improved the quality of our manuscript, and we hope that we now have addressed all concerns from all reviewers.

Below we have pasted all the remaining points from the reviewers in black and color coded our replies in red.

We have also included two “diff-files” highlighting all changes made. These are automatically generated and we believe them to be correct.

In these files we would like to point out that the latexdiff software failed to regenerate the caption of Table S2 and Fig. S12 correctly, which has been slightly modified as well. We have highlighted these changes manually, however the manual changes look visually different from those made by latexdiff. The “final” version of these tables/figures are available in the submitted SI.

Furthermore, Table S3 and S4 have been visually changed, but not the content. This is too complex to highlight in color code and thus only mentioned here. Essentially a few linebreaks were included to make the last column narrower.

In addition there are some minor changes not prompted by the reviewers, these are either grammatical or of a pure “LaTeX-formatting” nature.

We have also added a new reference (new Ref. 39 in the main paper) that was not available until now, this addition is not highlighted in the diff file. In short this reference describes the spectrometer used for these experiments.

Under each question/comment we have a section labeled **Changes**, this is to highlight what parts of the manuscript have been changed in relation to this question. Some questions are quite broad and span several aspects. We have then highlighted the most relevant changes to the manuscript, but we see that other changes can also be relevant for these questions. Creating an exhaustive list is non-trivial for these questions.

For simplicity we have ensured that each question/comment always starts on the top of a new page.

On behalf of all the authors,

Johan Söderström

REVIEWER COMMENTS

Reviewer #1 (Remarks to the Author):

The authors have provided, in their detailed response, sufficient evidence that the observed process has general applicability to solvated cationic and anionic species, and they have partially improved the manuscript accordingly. I consider the article potentially publishable in Nat. Comm., but the presentation and discussion require further revision.

We are happy that reviewer #1 finds that our manuscript has improved, and we hope that with our replies below the reviewer will find the manuscript ready to be published.

Following the reviewer's advice we have paid extra attention to the presentation and discussion in this revised version.

Changes:

- Detailed below.

I would appreciate if some of the discussion concerning the general applicability of the findings were included in the main text, rather than being confined to the response to reviewers. For example, it should be clarified under what conditions the features of the central ion and the solvation shell can be considered sufficiently isolated, and it should be emphasized that the method applies not only to cationic but also to anionic species.

We have added the results of both Cu^{2+} and F^- into the SI to highlight that IRD appears to be a general method, also for transition metal ions as well as for anions in water. A paragraph related to this was also added to the main paper.

We have been debating among the authors on how to interpret the question related to the question "*under what conditions the features of the central ion and the solvation shell can be considered sufficiently isolated*". Below we answer this in two different ways, first from a spectral point of view and secondly from the view of overlap of orbitals.

Spectral overlap: Concerning under what conditions the features of the central ion and the solvation shell can be considered sufficiently isolated, we don't think there is a simple answer to this. For the Na^+ and Mg^{2+} cases, the two types of features are energetically well separated. For the Cu^{2+} and F^- cases they partly overlap - especially visible for the F^- case. But the features of the central ion and the solvation shell can still be readily identified in the spectra. With theoretical support, it may well be possible to disentangle even more complex cases, which however is beyond the scope of this paper.

Orbital overlap: We have not further discussed when the "*features of the central ion and the solvation shell can be considered sufficiently isolated*" in this circumstance, as this would contain too much speculation at present. In our manuscript we have already mentioned that a non-zero overlap between the solute and solvent orbitals is required in order to observe IRD, which at least partially answers this.

Changes:

- SI: With the inclusion of Cu^{2+} and F^- IRD in the SI the reader can see how the spectral overlap between IRD and intra-atomic decay changes the results. See SI:
- SI: Line 40 - 42 Relates to the inclusion of experimental data from Cu^{2+} and F^-
- SI: Line 60 - 76 Relates to the inclusion of experimental data from Cu^{2+} and F^-
- SI: Fig S2 Relates to the inclusion of experimental data from Cu^{2+} and F^-
- Lines 387 - 399 in the main manuscript relates to the inclusion of experimental data from Cu^{2+} and F^-

Regarding the novelty of the work, which reviewer 4 has questioned, I would frame it as follows. There is no doubt that such a process exists, as it is consistent with quantum mechanics and has been observed in MOn species and in similar radiative and non-radiative probe studies. The novelty of the present work lies in demonstrating that the process is detectable in relatively weakly bound systems even at synchrotron facilities, and that the resulting signal contains sufficient structure to potentially yield more detailed insight. Otherwise, stating that the intensity of one-electron transitions is closely related to the overlap of the involved orbitals is merely a restatement of the locality of the dipole operator, which is a trivial observation. Moreover, the authors stress that this is the first such observation for a liquid system. I see no compelling reason why MnO₄⁻ or CrO₄²⁻ ions in solution should behave fundamentally differently than in solids, as studied decades ago. Therefore, the claimed novelty should focus on solvation-shell-specific aspects rather than the aggregation state.

We do agree with the reviewer that the focus of the manuscript should be on solvation-shell-specific aspects, as is apparent even in the title of the paper "*Providing an insider's view of the solvation shell by Intermolecular Radiative Decay*". That has always been our goal and we do believe that the focus is on this aspect. Here we believe that this paragraph is not related to any change needed, but rather an academic discussion by the reviewer thus we do not change anything in our manuscript based on this paragraph.

Furthermore, we can not agree with the statement "[...] restatement of the locality of the dipole operator, which is a trivial observation". The dipole operator is not localized, but rather the core holes are localized to a very large extent.

It is (or was) not obvious that the small overlap (or small one-center population) which, multiplied with the strong local transition dipole moment, should dominate over the off-center transitions with large population multiplied by weak (cross-transition) dipole moment. Here we confirm this, and so restore also for IRD the very useful one-center picture that has been used in interpretation of XES emission for decades.

Changes:

- Line 115-135 relates to this comment
- Line 176 - 184 relates to this comment

Another point concerns the repeated distinction between intra- and intermolecular transitions. In this context, that distinction is artificial, since there is no well-defined threshold for interaction strength or bond distance that separates the two regimes. As such, the novelty claims must be carefully reformulated; my suggestion above may serve as a starting point.

Regarding the distinction between intra- and intermolecular transitions, we agree with the reviewer that this can be non-trivial, as the overlap is never zero.

Using the example of MnO_4^- or CrO_4^{2-} ions in solution, we agree with the reviewer that these should not behave fundamentally differently than in solids. However, we would denote the transitions involving a Mn or Cr core-hole and the O atoms of the polyatomic ions as intra-molecular, whereas we would denote transitions involving a Mn or Cr core-hole and water molecules in the solvation shell as inter-molecular. In the case of Na and Mg presented in the manuscript, this distinction is based on the character of the LCAO expansion of the involved molecular orbitals, presented in Tables 1 and S1.

We also agree that it is by far better to focus this paper on the solvation-shell specific aspects. We believe that by the change suggested by reviewer #4 (see below) this is now implemented. In short this means re-writing a paragraph related to this distinction and moving it to the introduction.

Furthermore, we looked through our manuscript for the word “intra” and this appears in one place in the manuscript, namely in the revised paragraph suggested by reviewer #4 (see below). It does not appear in the SI at all. This was however discussed at length in our previous reply which could have led to the feeling that this was discussed at length in our manuscript.

Changes:

- Line 115 - 135 relates to this aspect
- Line 176 - 184 relates to this aspect

In the discussion of orbital composition and charge distribution, and throughout the article, the authors use language that suggests theoretical confirmation of the mechanism. Given that there is no plausible alternative explanation, this is not a confirmation but rather an interpretation. In my view, the explanation is rather straightforward, and I question the need for such detailed discussion in the main text. While this is a matter of taste, the “confirmation” narrative should be avoided.

This type of process is not previously observed or predicted in this energy regime, and we note that reviewer #2 was in his/her previous review as well as in the present reply sceptical to even observing IRD in these systems. We thus believe that for a wider audience, it is necessary to spend some time discussing the underlying mechanisms in some detail (why IRD is even possible). Once this decay has been observed and explained (as done in this paper) we do agree that the explanation is rather straightforward.

Regarding the underlying mechanisms, we agree with the reviewer that there is no plausible alternative explanation than the water molecules being involved in the radiative decay. We do not think that the explanation is completely straightforward: The one-center nature of X-ray emission was suggested already in the late 60'ies and one of the present authors investigated it in detail over a large body of molecules as early as in the 70'ies. It has been used numerous times since then. It generally holds for the main case when the emitting molecular orbital in covalently bonded systems has population around the atom with the core hole. This is not the case for IRD, and the question arises if cross-transition contributions outscore the one-center ones to the total dipole moment (see also previous comment). The results or “interpretation” presented in this manuscript is that it does not. Considering the utility of the one-center rule for XES in assigning local densities of certain symmetries we believe that this is a non-trivial as well as useful outcome of the paper.

Regarding the narrative, we agree with the reviewer that the theory presented is not a confirmation but rather an interpretation. This is for example indicated by the first subsection of the “Results and discussion” section having the title “IRD: Observation and interpretation”. To further clarify this, we have made a number of changes to key phrases in the manuscript. These are listed below and also clearly seen in the diff-file.

Abstract: “analyze” replaced by “interpret”

End of “The IRD mechanism” section: “We conclude...” replaced by “Based on the theory presented above, our interpretation is...”

Summary and conclusions: “*The transitions observed in the IRD spectra are shown to get intensity from hybridization between valence orbitals on water molecules and the occupied orbitals on the metal cation.*”

is now replaced by

“*The transitions observed in the IRD spectra are interpreted as getting intensity from hybridization between valence orbitals on water molecules and the occupied orbitals on the metal cation.*”

Changes:

- Line 168
- Line 299
- Line 400

The conclusion statement: “The IRD spectra are shown to reflect fundamental properties of the solvation shell, e.g., its radius, composition including possible ion pairing, electronic structure, and orientational disorder.” should be removed. These aspects are not addressed in the main text, and the supporting information only hints at, without providing conclusive analysis, that such properties might be inferable in future studies. Moreover, other well-established techniques can probe these properties more directly. As I noted in my initial review, merely observing a spectral shift does not equate to gaining insight. The energy shift is an integral quantity that depends on many variables, and drawing specific conclusions would require extensive correlation studies supported by well-resolved spectral features.

We believe that the IRD spectra can be shown to reflect fundamental properties of the solvation shell, e.g., its radius, composition including possible ion pairing, electronic structure, and orientational disorder. The present work has given such indications, but much of this has to await further studies covering a wider set of systems of different character. We have thus, as suggested by the reviewer, removed the statement from the paper as these discussions are based on theoretical calculations and not experimental findings at present.

Changes

- Line 409 - 411

In some places, the structure of sentences and paragraphs does not follow logical flow, implying unwarranted connections between unrelated ideas. For example: Line 68: "Electronic structure is responsible for the chemical interaction" is too colloquial for scientific writing. Moreover, the second part of the sentence does not logically follow from the first.

We agree that this should be changed and we have re-written this sentence.

The old sentence read:

"As the electronic structure is responsible for the chemical interaction of the solute with the neighboring solvent, an experimental tool capable of selective probing would be very valuable."

While the new reads:

"An experimental probe of the electronic interaction between the solute with the neighboring solvent could thus lead to an increased understanding of this interaction."

Changes:

- Lines 67 - 71 relates to this comment

Lines 95–100: The authors state that the solvation shell of Na^+ and Mg^{2+} consists of six water molecules, followed by an illogical implication about the energetic position of radiative decay features relative to $K\alpha$.

In the isolated Na^+ and Mg^{2+} ions the K_α decay corresponds to the highest energetic photon decay possible. Should photons with an even higher energy be observed they must originate from interaction with the surrounding.

IRD decay can indeed be observed at lower emission energies (see e.g. F^-). However, the statement in the manuscript is correct as it is written for the case of Na^+ and Mg^{2+} . We do not see how the text in its current form has any implication between the number of molecules in the solvation shell and the energy position of the IRD peaks. We have not made any changes to the text since we do not see how to improve the readability of this statement.

Changes:

- None, we note however that the line numbering is naturally different in the new version.

Additionally, the final two paragraphs on page 3 should be swapped: first introduce the general scope of the research, then discuss the details. As currently written, the logical flow of that section is unclear.

The reviewer is correct in this and we have swapped the order of these paragraphs.

Changes:

- Lines 91 - 99 and 127 - 135

I also do not share the authors' enthusiasm for repeatedly describing the probe as "from within." What would the alternative be, probing "from outside"? Would that yield fundamentally different information? What these types of processes probe is the local electronic structure, again due to the locality of the transition operator.

There are other conceivable possibilities to probe the solvation shell. One is the differential approach used in Ref. 37 and discussed in the manuscript. Another possibility could have been selective excitation of the solvation shell water molecules, but in the present cases there is no such unique solvation-shell-related spectral feature in the X-ray absorption spectrum. We used the expression "from within" as a way of stressing that we instead excite/ionize the solute ion inside the solvation shell.

We note that the word "within" (in this context) appears once in the abstract, twice in the main manuscript and not at all in the SI. In order to reduce the repeatedness we have changed the paragraph that used to read

"These results for hydrated Na^+ and Mg^{2+} ions demonstrate the potential of IRD as a more generally applicable method to selectively probe the solvation shell from within, thereby opening new possibilities to understand how the solvation shell affects the chemical properties of solutes."

Into

"These results for hydrated Na^+ and Mg^{2+} ions demonstrate the potential of IRD as a more generally applicable method where it is possible to use the solvated ions to selectively probe the solvation shell around each species, thereby opening new possibilities to understand how the solvation shell affects the chemical properties of solutes."

Changes:

- Lines 413 - 414

It should be explicitly stated that the “HF calculations” are in fact DeltaSCF HF calculations in the main text.

We have implemented several changes related to this comment. Specifically related to this statement we have included the sentence

“The molecular orbitals were optimized for the ground, core-hole and final states.” in the SI, however numerous small changes have been introduced to clarify the theoretical modeling at various places throughout the manuscript.

Changes:

- What models have been used is now specified in several places in the main manuscript as well as in the SI - specifically in most figure/table captions.
- Line 516 relates to this comment
- SI: Line 215 - 216 relates to this comment

Reviewer #2 (Remarks to the Author):

I appreciate the author's thoughtful response to the original review. Unfortunately, the changes that have been made to the manuscript, especially regarding clarifying the computational details, are still insufficient to merit publication in the manuscript's current form. More specific comments can be found below:

We thank the reviewer for taking his/her time to read this revised manuscript and also acknowledge that (s)he sees that the present revision is based on a thoughtful response.

Regarding the computational details this is discussed and answered in connection with question #3 below.

1. Fig. 4 has been visually improved. However, the author's conclusion that IRD is a powerful technique to examine the electronic structure of the surrounding solvent still seems dubious. For Na, it is difficult to visually tell a huge difference between the bulk liquid and the experimental IRD spectrum. Furthermore, the vertical lines from reference 14 and theoretical calculations in this manuscript, do not seem to show good agreement with the observed experimental spectrum. For Mg, there does seem to be a bigger difference between the bulk liquid and IRD spectrum, but once again there seems to be poor agreement with the theoretical results making interpretation difficult.

Here it is clear that reviewer #2 is of the opposite opinion as compared to reviewers #1, #3 and #4.

In the case of Na, not only is there a shift in the energy-positions of the IRD peaks as compared to the photoemission spectrum of bulk water, but the peak that would naively correspond to the $1b_2$ peak is not observed at all. So in this matter we do not agree with the reviewer.

Interestingly the Mg-spectrum does contain three peaks, so in that aspect it better resembles the photoemission spectrum of bulk water, but the relative peak intensities are vastly different. Already here we can observe that the interaction with the first solvation shell is different for the Na and Mg ions, which is the strength of IRD.

The reviewer is indeed correct in his/her statement that the energy vertical lines from Ref. 14 do not agree perfectly in energy, nor with our experimental finding nor our theoretical calculations. The fact that theoretical calculations can differ from experimental findings in dynamic complex systems is not surprising.

Furthermore the reviewer points out that theoretical calculations from our manuscript are not perfect in agreement with the experimental data. As stated in our manuscript the theoretical calculations are not from all possible molecular configurations of the solvation shell, but rather from "a representative cluster geometry derived from molecular dynamics simulations". A complete theoretical calculation coupling MD to HF calculations will clearly give different results, however such calculations are too expensive to perform at present. This is already elaborated on in our manuscript.

We would also like to remind the reviewer that the studied IRD transitions are two orders of magnitude weaker than "normal" X-ray transitions, thereby becoming much more sensitive to the character of wave functions, density functionals and to the computational scheme applied. We remind the reviewer that our work combines wave function theory, density functional theory and classical molecular dynamics which by itself represents an ambitious level of approach. The two latter are combined to allow for bigger clusters and dynamics studies. However, due to computational restraints, spectra from full sampling of long time trajectories (the so-called integrated approach) could not be fully accomplished. Rather we had to resort to selected clusters that appear predominantly in the MD. This together with the weakness of the transitions (see above) are probably the causes of discrepancies.

Changes:

- None

2. The author's make a seemingly strange or incorrect comment regarding the timescale of the IRD process in response to my original comments as well as in response to a comment from reviewer 1. Specifically, they state that the IRD process probes the core-hole lifetime (on the order of a few fs), but that the IRD process occurs on a timescale of IRD is on the order of ps. These statements seem contradictory. The timescale of the core-hole lifetime should be given by the timescale for the processes that lead to core-hole decay. Therefore, if the core-hole lifetime is on the order of fs, then the core-hole should be decaying via a process that occurs on the fs timescale (such as Auger-Meitner decay or some other more complicated process such as ICD or ETMD). If IRD occurs on the ps timescale, then the core-hole would need to exist on the timescale of ps, which is not the case. Therefore, unfortunately, this reviewer still seems somewhat skeptical about how IRD is physically feasible in this system based on these comments.

We would first like to point out that IRD is clearly physically feasible in these and other systems, since we do clearly observe IRD in Na^+ , Mg^{2+} as well as in Cu^{2+} and F^- as shown in the SI. Furthermore, we would like to point out the reply of reviewer #1 above where (s)he states that “*There is no doubt that such a process exists*” and “*Given that there is no plausible alternative explanation*”.

We note that the discussion related to timescales arose from a question posed by the reviewer in the first round and we did our best to answer this in terms of timescales in the reply. We also note that timescales are not discussed in the manuscript. However, we realize that our reply to the reviewer's question about timescales has led to further misunderstandings about this concept, and we will here try to rectify this.

Let's turn our attention to the sentence “*If IRD occurs on the ps timescale, then the core-hole would need to exist on the timescale of ps, which is not the case*”. We believe that this sentence can be at the heart of the misunderstanding of timescales and decay rates. Here we would, again, like to point out that the different core hole decay paths do **not** occur on different timescales – they all probe the same timescale, set by the core hole lifetime (~2 fs in these cases). However, the different decays occur with different decay rates (and thus different decay probabilities).

The timescales quoted in our previous reply were all inferred from peak intensities, and the reason we did this was to answer the specific question related to timescales. We however stress that in our previous reply we specifically wrote “*However, we would like to stress that this does not mean that IRD probes a longer time scale than Auger-Meitner. Instead, all the core-hole decay processes probe the same time scale, set by the 1s core-hole lifetimes*”. In other words, the core hole decays through a multitude of decay paths, each with a different rate, R_i . The probability P_i of a particular decay, i , is given by:

$$P_i = \frac{R_i}{R_{Tot}}$$

where the total decay rate is the sum of all rates, $R_{Tot} = \sum_i R_i$. This in turn defines the lifetime of the core hole as $\tau = \frac{1}{R_{Tot}}$. The lifetime of the core holes in our manuscript is on the order of 2 fs.

It may be appealing to convert these different decay rates, R_i , into different lifetimes (or timescales as per the nomenclature of the reviewer), but this is not correct. The lifetime of the corehole is related to the sum of **all** decay rates (R_{Tot}), and there is no specific timescale for IRD, nor for any other decay path.

Peaks with relatively low intensity thus correspond to decay paths with a low decay rate (which does **not** imply “long timescales”) and peaks with relatively high intensity correspond to a higher decay rate (which does **not** imply “short timescales”). The core hole will decay through

all possible decay paths, albeit with different probabilities, P_i . These probabilities, P_i , are directly related to the decay rate R_i as shown in the equation above and the effect of this is observed as different intensities of the peaks in our spectra.

The concept of decay rates and lifetimes is further discussed in several textbooks, e.g. Ref. [1].

Summarizing, the core hole will decay through all possible decay paths on a timescale, set by the core hole lifetime (~ 2 fs in the present cases). Some decay paths with a higher probability (e.g. the Auger-Meitner decay, an electron emission signal not observable in our X-ray photon spectra) and others with a lower probability (e.g. the local radiative K_α decay, which is observed in our X-ray photon spectra). The IRD decay has an even lower probability than the K_α decay, however still sufficient to be observed, as shown in both our measurements and theoretical calculations.

We hope that this answer has cleared up any confusion related to both timescales and why IRD is indeed observed.

Changes:

- None

3. The only comments to the types of calculations in the analysis and discussion are still only with respect to the Hartree-Fock (HF) approximation. However, the calculation details are still with respect to RAS calculations. Therefore, the same confusion from the original form of the manuscript remains; there is a complete discrepancy between the calculation details and the results presented in the manuscript. What is needed is very concrete statements, such as the results in Figure 1 are obtained from explicitly this type of calculation etc. etc.

This is a very valid point and also related to a point brought up by reviewer #1. We have addressed this in several places of the manuscript and we do believe that this is now much improved. The changes related to both captions but also changes to the text in the manuscript and SI.

Changes:

- Several changes throughout the manuscript and SI. The caption of Fig. 4 was not changed since it is stated already in Fig. 2
- Line 516 relates to this comment
- SI: Line 215 - 216 relates to this comment
- SI: Fig. S12 Caption updated, relates to this comment

4. Though the authors comment that a change has been made, the phrase radiative decay rates is still used in the computational details section.

This is absolutely true and we do not understand how this could be missed by us, we thank the reviewer for (again) pointing this out. We have now re-implemented this change. However, as before we note that the two concepts are related.

Changes:

- Line 475 relates to this comment

Reviewer #4 (Remarks to the Author):

The authors have addressed most of the comments, but I would like to see two important changes to the modified manuscript. If the authors make these changes, I support publication.

We would like to thank the reviewer for very insightful comments in his/her previous reply as well as here. Furthermore we would like to thank the reviewer for even suggesting a paragraph – we adapted this paragraph more or less as-is into the introduction of our manuscript. We do believe that this change made our manuscript even better. With this change we believe that we have addressed the two points raised by reviewer #4 below.

Changes (this relates to both comments by reviewer #4):

- Line 115-135
- Line 176 - 184

1) The paragraph “Radiative transitions involving a core hole on one atom and a valence orbital mainly located on another atom....probing chemically and biologically relevant systems in an aqueous environment.” needs to be moved to the introduction, as the distinction is important background information for the reader and needs to be addressed early on.

2) Some additional clarification/correction in this paragraph is needed. It is not true that the inter atomic transitions previously observed only occur in covalently bonded systems, they also occur for ionic bonding, such as in MnO for example. The authors should also add the word ‘inter atomic’ to the description to clarify the distinction between intra molecular (but inter atomic) and inter molecular. This is important as regular XES is always intra molecular (and generally intra atomic).

After moving the text to the introduction, the paragraph could be modified to something like this (plus consistency of reference numbers and changes to the flow of the narrative):

“Radiative transitions involving a core hole on one atom and a valence orbital mainly located on a neighboring atom have been well known for a long time. In transition metal-oxygen ligand systems, such as MnO_4^- , CrO_4^{2-} and VO_4^{3-} , inter-atomic radiative decay has been observed from orbitals primarily located on the ligand oxygen atoms into core-holes centered on the metal ion, see, e.g., [18–23]. Compared to the ions studied here, which form much weaker bonds with the surrounding water, the systems studied in Refs. [18–23] are held together more strongly either by covalent bonds in molecules, or ionic/covalent bonds in the crystalline systems. These inter-atomic transitions observed for the strongly bonded molecular systems are still within the same molecule and can therefore be characterized as intra-molecular. IRD, on the other hand, is inter-molecular, opening up the possibility of probing chemically and biologically relevant systems in an aqueous environment.”

References

- [1] J. Stöhr, “The Nature of X-Rays and Their Interactions with Matter,” vol. 288, 2023, doi: 10.1007/978-3-031-20744-0.

We would again like to take the opportunity to take the time to extend our thanks to the reviewers for taking the time to carefully read both the manuscript and our replies to their comments. This has certainly improved our manuscript and we have enjoyed the academic discussion in this review process.

At present there are three questions from reviewer #2 remaining and we would like to address them here.

1. As far as I can tell, the main text does not contain any results using the RAS calculations. Consequently, the computational details associated with the RAS calculations should be moved to the corresponding section of the SI that uses RAS. Instead, the computational details section in the main text should focus on the data presented in the main text to not confuse the reader.

As how interpret the instructions from from the editor (specified in the Author Checklist) this information should be available in the Methods section of the main manuscript. Thus at present we have not changed anything related to this comment.

Changes: None

2. The details with respect to the calculation of the XES spectra in terms of the HF approximation should be more clearly emphasized and made into its own paragraph. These calculations are the main ones utilized in the main text and therefore need to be the most obvious and clearly stated.

Changes: We have added a paragraph and references to this into the methods section.

3. On lines 312 and 319, the authors should specify which sections of the SI they're referring to.

We have updated the manuscript at all places where we refer to the SI by including the relevant sections we refer to.

Changes: All references to the SI now includes the relevant section to look into.